# Enemy is Inside: Alleviating VAE's Overestimation in Unsupervised OOD Detection

## Abstract

Deep generative models (DGMs) aim at characterizing the distribution of the training set by maximizing the marginal likelihood of inputs in an unsupervised manner, making them a promising option for unsupervised out-of-distribution (OOD) detection. However, recent works have reported that DGMs often assign higher likelihoods to OOD data than in-distribution (ID) data, *i.e.*, ***overestimation***, leading to their failures in OOD detection. Although several pioneer works have tried to analyze this phenomenon, and some VAE-based methods have also attempted to alleviate this issue by modifying their score functions for OOD detection, the root cause of the *overestimation* in VAE has never been revealed to our best knowledge. To fill this gap, this paper will provide a thorough theoretical analysis on the *overestimation* issue of VAE, and reveal that this phenomenon arises from two Inside-Enemy aspects: 1) the improper design of prior distribution; 2) the gap of dataset entropies between ID and OOD datasets. Based on these findings, we propose a novel score function to **A**lleviate **V**AE's **O**verestimation **I**n unsupervised OOD **D**etection, named **"AVOID"**, which contains two novel techniques, specifically post-hoc prior and dataset entropy calibration. Experimental results verify our analysis, demonstrating that the proposed method is effective in alleviating *overestimation* and improving unsupervised OOD detection performance.

## 1 Introduction

The detection of out-of-distribution (OOD) data, *i.e.*, identifying data that differ from the in-distribution (ID) training set, is crucial for ensuring the reliability and safety of real-world applications [1, 2, 3, 4]. While the most commonly used OOD detection methods rely on supervised classifiers [5, 6, 7, 8, 9, 10, 11], which require labeled data, the focus of this paper is on designing an unsupervised OOD detector. **Unsupervised OOD detection** refers to the task of designing a detector, based solely on the unlabeled training data, that can determine whether an input is ID or OOD [12, 13, 14, 15, 16, 17, 18]. This unsupervised approach is more practical for real-world scenarios where the data lack labels.

Deep generative models (DGMs) are a highly attractive option for unsupervised OOD detection. DGMs, mainly including the auto-regressive model [19, 20], flow model [21, 22], diffusion model [23], generative adversarial network [24], and variational autoencoder (VAE) [25], are designed to model the distribution of the training set by explicitly or implicitly maximizing the likelihood estimation of $p(\boldsymbol{x})$ for its input $\boldsymbol{x}$ without category label supervision or additional OOD auxiliary data. They have achieved great successes in a wide range of applications, such as image and text generation. Since generative models are promising at modeling the distribution of the training set, they could be seen as an ideal unsupervised OOD detector, where the likelihood of the unseen OOD data output by the model should be lower than that of the in-distribution data.

Submitted to 37th Conference on Neural Information Processing Systems (NeurIPS 2023). Do not distribute.

Unfortunately, developing a flawless unsupervised OOD detector using DGMs is not as easy as it seems to be. Recent experiments have revealed a counterfactual phenomenon that directly applying the likelihood of generative models as an OOD detector can result in **overestimation**, *i.e.*, **DGMs assign higher likelihoods to OOD data than ID data** [12, 13, 17, 18]. For instance, a generative model trained on the FashionMNIST dataset could assign higher likelihoods to data from the MNIST dataset (OOD) than data from the FashionMNIST dataset (ID), as shown in Figure 6(a). Since OOD detection can be viewed as a verification of whether a generative model has learned to model the distribution of the training set accurately, the counterfactual phenomenon of *overestimation* not only poses challenges to unsupervised OOD detection but also raises doubts about the generative model's fundamental ability in modeling the data distribution. Therefore, it highlights the need for developing more effective methods for unsupervised OOD detection and, more importantly, a more thorough understanding of the reasons behind the *overestimation* in deep generative models.

To develop more effective methods for unsupervised OOD detection, some approaches have modified the likelihood to new score functions based on empirical assumptions, such as low- and high-level features' consistency [17, 18] and ensemble approaches [26]. While these methods, particularly the VAE-based methods [18], have achieved state-of-the-art (SOTA) performance in unsupervised OOD detection, none of them provides a clear explanation for the *overestimation* issue. To gain insight into the *overestimation* issue in generative models, pioneering works have shown that the *overestimation* issue could arise from the intrinsic model curvature brought by the invertible architecture in flow models [27]. However, in contrast to the exact marginal likelihood estimation used in flow and auto-regressive models, VAE utilizes a lower bound of the likelihood, making it difficult to analyze. Overall, the reasons behind the *overestimation* issue of VAE are still not fully understood.

In this paper, we try to address the research gap by providing a theoretical analysis of VAE's *overestimation* in unsupervised OOD detection. Our contributions can be summarized as follows:

1. Through theoretical analyses, we are the first to identify two factors that cause the *overestimation* issue of VAE: 1) the improper design of prior distribution; 2) the intrinsic gap of dataset entropies between ID and OOD datasets;
2. Focused on these two discovered factors, we propose a new score function, named **"AVOID"**, to alleviate the *overestimation* issue from two aspects: 1) post-hoc prior for the improper design of prior distribution; 2) dataset entropy calibration for the gap of dataset entropies;
3. Extensive experiments demonstrate that our method can effectively improve the performance of VAE-based methods on unsupervised OOD detection, with theoretical guarantee.

## 2 Preliminaries

### 2.1 Unsupervised Out-of-distribution Detection

In this part, we will first give a problem statement of OOD detection and then we will introduce the detailed setup for applying unsupervised OOD detection.

**Problem statement.** While deploying a machine learning system, it is possible to encounter inputs from unknown distributions that are semantically and/or statistically different from the training data, and such inputs are referred to as OOD data. Processing OOD data could potentially introduce critical errors that compromise the safety of the system [1]. Thus, the OOD detection task is to identify these OOD data, which could be seen as a binary classification task: determining whether an input $x$ is more likely ID or OOD. It could be formalized as a level-set estimation:

$$x = \begin{cases} \text{ID}, & \text{if} \quad \mathcal{S}(x) > \lambda, \\ \text{OOD}, & \text{if} \quad \mathcal{S}(x) \leq \lambda, \end{cases} \tag{1}$$

where $\mathcal{S}(x)$ denotes the score function, *i.e.*, **OOD detector**, and the threshold $\lambda$ is commonly chosen to make a high fraction (*e.g.*, 95%) of ID data is correctly classified [9]. In conclusion, OOD detection aims at designing the $\mathcal{S}(x)$ that could assign higher scores to ID data samples than OOD ones.

**Setup.** Denoting the input space with $\mathcal{X}$, an *unlabeled* training dataset $\mathcal{D}_{\text{train}} = \{x_i\}_{i=1}^{N}$ containing of $N$ data points can be obtained by sampling *i.i.d.* from a data distribution $\mathcal{P}_{\mathcal{X}}$. Typically, we treat the $\mathcal{P}_{\mathcal{X}}$ as $p_{\text{id}}$, which represents the in-distribution (ID) [17, 27]. With this *unlabeled* training set, unsupervised OOD detection is to design a score function $\mathcal{S}(x)$ that can determine whether an input is ID or OOD. This is different from supervised OOD detection, which typically leverages a classifier that is trained on labeled data [4, 7, 9]. We provide a detailed discussion in Appendix A.

## 2.2 VAE-based Unsupervised OOD Detection

DGMs could be an ideal choice for unsupervised OOD detection because the estimated marginal likelihood $p_\theta(x)$ can be naturally used as the score function $\mathcal{S}(x)$. Among DGMs, VAE can offer great flexibility and strong representation ability [28], leading to a series of unsupervised OOD detection methods based on VAE that have achieved SOTA performance [17, 18]. Specifically, VAE estimates the marginal likelihood by training with the variational evidence lower bound (ELBO), *i.e.*,

$$\text{ELBO}(x) = \mathbb{E}_{q_\phi(z|x)}[\log p_\theta(x|z)] - D_{\text{KL}}(q_\phi(z|x)||p(z)), \tag{2}$$

where the posterior $q_\phi(z|x)$ is modeled by an encoder, the reconstruction likelihood $p_\theta(x|z)$ is modeled by a decoder, and the prior $p(z)$ is set as a Gaussian distribution $\mathcal{N}(0, I)$. After well training the VAE, $\text{ELBO}(x)$ is an estimation of the $p(x)$, which could be directly seen as the score function $\mathcal{S}(x)$ to do OOD detection. But the VAE would suffer from the *overestimation* issue, which will be introduced in the next section. More details and **Related Work** can be seen in Appendix B.

# 3   Analysis of VAE's *overestimation* in Unsupervised OOD Detection

We will first conduct an analysis to identify the factors contributing to VAE's *overestimation*, *i.e.*, the improper design of prior distribution and the gap between ID and OOD datasets' entropies. Subsequently, we will give a deeper analysis of the first factor to have a better understanding.

## 3.1   Identifying Factors of VAE's *Overestimation* Issue

Following the common analysis procedure [27], an ideal score function $\mathcal{S}(x)$ that could achieve good OOD detection performance is expected to have the following property for any OOD dataset:

$$\mathcal{G} = \mathbb{E}_{x \sim p_{\text{id}}(x)}[\mathcal{S}(x)] - \mathbb{E}_{x \sim p_{\text{ood}}(x)}[\mathcal{S}(x)] > 0, \tag{3}$$

where $p_{\text{id}}(x)$ and $p_{\text{ood}}(x)$ denote the true distribution of the ID and OOD dataset, respectively. A larger gap between these two expectation terms can usually lead to better OOD detection performance.

Using the $\text{ELBO}(x)$ as the score function $\mathcal{S}(x)$, we could give a formal definition of the repeatedly reported VAE's *overestimation* issue in the context of unsupervised OOD detection [12, 13, 17, 18].

**Definition 1** (VAE's *overestimation* in unsupervised OOD Detection)**.** Assume we have a VAE trained on a training set and we use the $\text{ELBO}(x)$ as the score function to distinguish data points sampled *i.i.d.* from the in-distribution testing set ($p_{\text{id}}$) and an OOD dataset ($p_{\text{ood}}$). When

$$\mathcal{G} = \mathbb{E}_{x \sim p_{\text{id}}(x)}[\text{ELBO}(x)] - \mathbb{E}_{x \sim p_{\text{ood}}(x)}[\text{ELBO}(x)] \leq 0, \tag{4}$$

it is called VAE's *overestimation* in unsupervised OOD detection.

With a clear definition of *overestimation*, we could now investigate the underlying factors causing the *overestimation* in VAE. After well training a VAE, we could reformulate the expectation term of $\text{ELBO}(x)$ from the perspective of information theory [29] as:

$$\mathbb{E}_{x \sim p(x)}[\text{ELBO}(x)] = \mathbb{E}_{x \sim p(x)}[\mathbb{E}_{z \sim q_\phi(z|x)} \log p_\theta(x|z)] - \mathbb{E}_{x \sim p(x)}[D_{\text{KL}}(q_\phi(z|x)||p(z))]$$
$$= -\mathcal{H}_p(x) - D_{\text{KL}}(q(z)||p(z)), \tag{5}$$

because we have

$$\mathbb{E}_{x \sim p(x)}[\mathbb{E}_{z \sim q_\phi(z|x)} \log p_\theta(x|z)] = \mathcal{I}_q(x, z) + \mathbb{E}_{p(x)} \log p(x) = \mathcal{I}_q(x, z) - \mathcal{H}_p(x), \tag{6}$$
$$\mathbb{E}_{x \sim p(x)}[D_{\text{KL}}(q_\phi(z|x)||p(z))] = \mathcal{I}_q(x, z) + D_{\text{KL}}(q(z)||p(z)), \tag{7}$$

where the $\mathcal{I}_q(x, z)$ is mutual information between $x$ and $z$ and the $q(z)$ is the aggregated posterior distribution of the latent variables $z$, which is defined by $q(z) = \mathbb{E}_{x \sim p(x)} q_\phi(z|x)$. We leave the detailed definition and derivation in Appendix C.1. Thus, the gap $\mathcal{G}$ in Eq. (4) could be rewritten as

$$\mathcal{G} = [-\mathcal{H}_{p_{\text{id}}}(x) + \mathcal{H}_{p_{\text{ood}}}(x)] + [-D_{\text{KL}}(q_{\text{id}}(z)||p(z)) + D_{\text{KL}}(q_{\text{ood}}(z)||p(z))], \tag{8}$$

where the dataset entropy $\mathcal{H}_{p_{\text{id}}}(x)/\mathcal{H}_{p_{\text{ood}}}(x)$ is a constant that only depends on the true distribution of ID/OOD dataset; the prior $p(z)$ is typically set as a standard (multivariate) Gaussian distribution $\mathcal{N}(0, I)$ to enable reparameterization for efficient gradient descent optimization [25].

Through analyzing the most widely used criterion, specifically the expectation of ELBO reformulated in Eq. (8), for VAE-based unsupervised OOD detection, we find that there will be two potential factors that lead to the *overestimation* issue of VAE, *i.e.*, $\mathcal{G} \leq 0$:

**Factor I: The improper design of prior distribution** $p(\boldsymbol{z})$. Several studies have argued that the aggregated posterior distribution of latent variables $q(\boldsymbol{z})$ cannot always equal $\mathcal{N}(\boldsymbol{0}, \mathbf{I})$, particularly when the dataset exhibits intrinsic multimodality [28, 30, 31, 32]. In fact, when $q(\boldsymbol{z})$ is extremely close to $p(\boldsymbol{z})$, it is more likely to become trapped in a bad local optimum known as posterior collapse [33, 34, 35], *i.e.*, $q_\phi(\boldsymbol{z}|\boldsymbol{x}) \approx p(\boldsymbol{z})$, resulting in $q(\boldsymbol{z}) = \int_{\boldsymbol{x}} q_\phi(\boldsymbol{z}|\boldsymbol{x})p(\boldsymbol{x}) \approx \int_{\boldsymbol{x}} p(\boldsymbol{z})p(\boldsymbol{x}) = p(\boldsymbol{z})$. In this situation, the posterior $q_\phi(\boldsymbol{z}|\boldsymbol{x})$ becomes uninformative about the inputs. Thus, the value of $D_{\mathrm{KL}}(q_{\mathrm{id}}(\boldsymbol{z})||p(\boldsymbol{z}))$ could be overestimated, potentially contributing to $\mathcal{G} \leq 0$.

**Factor II: The gap between** $\mathcal{H}_{p_{\mathrm{id}}}(\boldsymbol{x})$ **and** $\mathcal{H}_{p_{\mathrm{ood}}}(\boldsymbol{x})$. Considering the dataset's statistics, such as the variance of pixel values, different datasets exhibit various levels of entropy. It is reasonable that a dataset containing images with richer low-level features and more diverse content is expected to have a higher entropy. As an example, the FashionMNIST dataset should possess higher entropy compared to the MNIST dataset. Therefore, when the entropy of the ID dataset is higher than that of an OOD dataset, the value of $-\mathcal{H}_{p_{\mathrm{id}}}(\boldsymbol{x}) + \mathcal{H}_{p_{\mathrm{ood}}}(\boldsymbol{x})$ is less than 0, potentially leading to *overestimation*.

## 3.2 More Analysis on Factor I

In this part, we will focus on addressing the following question: *when is the common design of the prior distribution proper, and when is it not?*

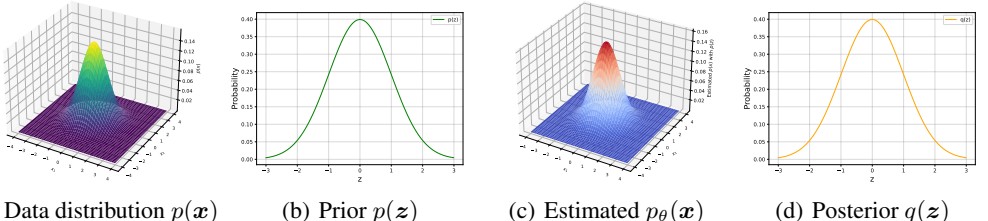

(a) Data distribution $p(\boldsymbol{x})$    (b) Prior $p(\boldsymbol{z})$    (c) Estimated $p_\theta(\boldsymbol{x})$    (d) Posterior $q(\boldsymbol{z})$

Figure 1: Visualization of modeling a single-modal data distribution with a linear VAE.

**When the design of prior is proper?** Assuming that we have a dataset consisting of $N$ data points $\{\boldsymbol{x}_i\}_{i=1}^N$, each of which is sampled from a given $d$-dimensional data distribution $p(\boldsymbol{x}) = \mathcal{N}(\boldsymbol{x}|\boldsymbol{0}, \boldsymbol{\Sigma}_{\mathbf{x}})$ as shown in Figure 1(a). Then we construct a linear VAE to estimate $p(\boldsymbol{x})$, formulated as:

$$
\begin{aligned}
p(\boldsymbol{z}) &= \mathcal{N}(\boldsymbol{z}|\boldsymbol{0}, \mathbf{I}) \\
q_\phi(\boldsymbol{z}|\boldsymbol{x}) &= \mathcal{N}(\boldsymbol{z}|\mathbf{A}\boldsymbol{x} + \mathbf{B}, \mathbf{C}) \\
p_\theta(\boldsymbol{x}|\boldsymbol{z}) &= \mathcal{N}(\boldsymbol{x}|\mathbf{E}\boldsymbol{z} + \mathbf{F}, \sigma^2\mathbf{I}),
\end{aligned}
\tag{9}
$$

where $\mathbf{A},\mathbf{B},\mathbf{C},\mathbf{D},\mathbf{E},\mathbf{F}$, and $\sigma$ are all learnable parameters and their optimal values can be obtained by the derivation in Appendix C.3. As the estimated distribution $p_\theta(\boldsymbol{x})$ depicted in Figure 1(c), we can find that the linear VAE with the optimal parameter values can accurately estimate the $p(\boldsymbol{x})$ through maximizing ELBO, *i.e.*, the *overestimation* issue is not present. In this case, Figures 1(b) and 1(d) indicate that the design of the prior distribution is proper, where the posterior $q(\boldsymbol{z})$ equals prior $p(\boldsymbol{z})$.

**When the design of prior is NOT proper?** Consider a more complex data distribution, *e.g.*, a mixture of Gaussians, $p(\boldsymbol{x}) = \sum_{k=1}^K \pi_k \mathcal{N}(\boldsymbol{x}|\boldsymbol{\mu}_k, \boldsymbol{\Sigma}_k), K = 2$ as shown in Figure 2(a), where $\pi_k = 1/K$ and $\sum_{k=1}^K \boldsymbol{\mu}_k = \boldsymbol{0}$. We construct a dataset consisting of $K \times N$ data points, obtained by sampling $N$ data samples $\{\boldsymbol{x}_i^{(k)}\}_{i=1,k=1}^{N,K}$ from each component Gaussian $\mathcal{N}(\boldsymbol{x}|\boldsymbol{\mu}_k, \boldsymbol{\Sigma}_k)$. The formulation of $p(\boldsymbol{z})$, $q_\phi(\boldsymbol{z}|\boldsymbol{x})$, and $p_\theta(\boldsymbol{x}|\boldsymbol{z})$ is consistent with those in Eq. (9). More details are in Appendix C.2.

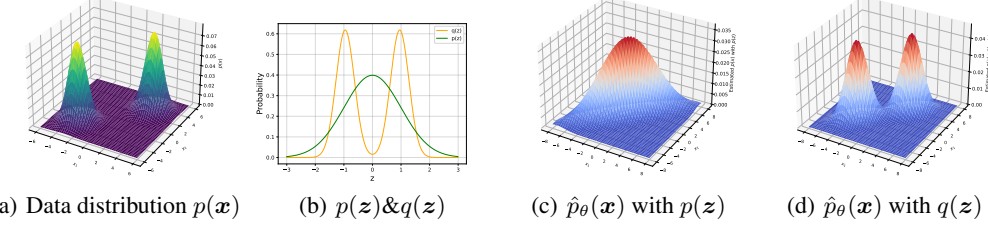

(a) Data distribution $p(\boldsymbol{x})$    (b) $p(\boldsymbol{z})\&q(\boldsymbol{z})$    (c) $\hat{p}_\theta(\boldsymbol{x})$ with $p(\boldsymbol{z})$    (d) $\hat{p}_\theta(\boldsymbol{x})$ with $q(\boldsymbol{z})$

Figure 2: Visualization of modeling a multi-modal data distribution with a linear VAE.

In what follows, we will provide a basic derivation outline for the linear VAE under the multi-modal case. We can first obtain the marginal likelihood $\hat{p}_\theta(\boldsymbol{x}; \mathbf{E}, \mathbf{F}, \sigma) = \int p_\theta(\boldsymbol{x}|\boldsymbol{z})p(\boldsymbol{z}) = \mathcal{N}(\boldsymbol{x}|\mathbf{F}, \mathbf{E}\mathbf{E}^\top +$

158    $\sigma^2\mathbf{I}$) with the strictly tighter importance sampling on ELBO [36], *i.e.*, learning the optimal generative
159    process. Then, the joint log-likelihood of the observed dataset $\{\boldsymbol{x}_i^{(k)}\}_{i=1,k=1}^{N,K}$ can be formulated as:

$$\mathcal{L} = \sum_{k=1}^{K}\sum_{i=1}^{N}\log\hat{p}_\theta(\boldsymbol{x}_i^{(k)}) = -\frac{KNd}{2}\log(2\pi) - \frac{KN}{2}\log det(\mathbf{M}) - \frac{KN}{2}tr[\mathbf{M}^{-1}\mathbf{S}], \quad (10)$$

160    where $\mathbf{M} = \mathbf{EE}^\top + \sigma^2\mathbf{I}$ and $\mathbf{S} = \frac{1}{KN}\sum_{k=1}^{K}\sum_{i=1}^{N}(\boldsymbol{x}_i^{(k)} - \mathbf{F})(\boldsymbol{x}_i^{(k)} - \mathbf{F})^\top$. After that, we could
161    explore the stationary points of parameters through the ELBO, which can be analytically written as:

$$\text{ELBO}(\boldsymbol{x}) = \overbrace{\mathbb{E}_{q_\phi(\boldsymbol{z}|\boldsymbol{x})}[\log p_\theta(\boldsymbol{x}|\boldsymbol{z})]}^{L_1} - \overbrace{D_{\text{KL}}[q_\phi(\boldsymbol{z}|\boldsymbol{x})||p(\boldsymbol{z})]}^{L_2}, \quad (11)$$

$$L_1 = \frac{1}{2\sigma^2}[-tr(\mathbf{ECE}^\top) - (\mathbf{EA}\boldsymbol{x} + \mathbf{EB})^\top(\mathbf{EA}\boldsymbol{x} + \mathbf{EB}) + 2\boldsymbol{x}^\top(\mathbf{EA}\boldsymbol{x} + \mathbf{EB}) - \boldsymbol{x}^\top\boldsymbol{x}] - \frac{d}{2}\log(2\pi\sigma^2),$$

$$L_2 = \frac{1}{2}[-\log det(\mathbf{C}) + (\mathbf{A}\boldsymbol{x} + \mathbf{B})^\top(\mathbf{A}\boldsymbol{x} + \mathbf{B}) + tr(\mathbf{C}) - 1].$$

162    The detailed derivation of parameter solutions in Eq. (10) and (11) can be found in Appendix C.4.

163    In conclusion of this case, Figure 2(b) illustrates that $q(\boldsymbol{z})$ is a multi-modal distribution instead of
164    $p(\boldsymbol{z}) = \mathcal{N}(\boldsymbol{z}|\boldsymbol{0}, \mathbf{I})$, *i.e.*, the design of the prior is not proper, which leads to *overestimation* as seen in
165    Figure 2(c). However, as analyzed in Factor I, we found that the *overestimation* issue is mitigated
166    when replacing $p(\boldsymbol{z})$ in the KL term of the ELBO with $q(\boldsymbol{z})$, which is shown in Figure 2(d).

167    **More empirical studies on the improper design of prior.** To extend to a more practical and
168    representative case, we used a 3-layer MLP to model $q_\phi(\boldsymbol{z}|\boldsymbol{x})$ and $p_\theta(\boldsymbol{x}|\boldsymbol{z})$ with $p(\boldsymbol{z}) = \mathcal{N}(\boldsymbol{0}, \mathbf{I})$ on
169    the same dataset of the above multi-modal case. Implementation details are provided in Appendix
170    C.5. After training, we observed that $q(\boldsymbol{z})$ still differs from $p(\boldsymbol{z})$, as shown in Figure 3(a). The ELBO
171    still suffers from *overestimation*, especially in the region near $(0, 0)$, as shown in Figure 3(b).

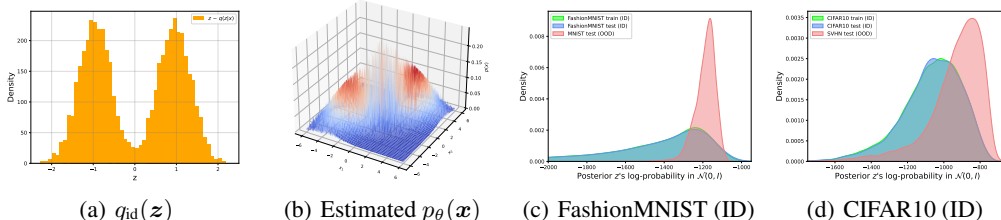

(a) $q_{\text{id}}(\boldsymbol{z})$      (b) Estimated $p_\theta(\boldsymbol{x})$      (c) FashionMNIST (ID)      (d) CIFAR10 (ID)

Figure 3: **(a)** and **(b)**: visualization of $q_{\text{id}}(\boldsymbol{z})$ and estimated $p(\boldsymbol{x})$ by ELBO on the multi-modal
data distribution with a non-linear deep VAE; **(c)** and **(d)**: the density plot of the log-probability of
posterior $\boldsymbol{z}$, *i.e.*, $\boldsymbol{z} \sim q_\phi(\boldsymbol{z}|\boldsymbol{x})$, in prior $\mathcal{N}(\boldsymbol{0}, \mathbf{I})$ on two dataset pairs.

172    Finally, we extend the analysis directly to high-dimensional image data. Since VAE trained on image
173    data needs to be equipped with a higher dimensional latent variable space, it is hard to visualize
174    directly. But please note that, if $q_{\text{id}}(\boldsymbol{z})$ is closer to $p(\boldsymbol{z}) = \mathcal{N}(\boldsymbol{0}, \mathbf{I})$, $\boldsymbol{z}_{\text{id}} \sim q_{\text{id}}(\boldsymbol{z})$ should occupy
175    the center of latent space $\mathcal{N}(\boldsymbol{0}, \mathbf{I})$ and $\boldsymbol{z}_{\text{ood}} \sim q_{\text{ood}}(\boldsymbol{z})$ should be pushed far from the center, leading
176    to $p(\boldsymbol{z}_{\text{id}})$ to be larger than $p(\boldsymbol{z}_{\text{ood}})$. However, surprisingly, we found this expected phenomenon
177    does not exist, as shown in Figure 3(c) and 3(d), where the experiments are on two dataset pairs,
178    Fashion-MNIST(ID)/MNIST(OOD) and CIFAR10(ID)/SVHN(OOD). This still suggests that the
179    prior $p(\boldsymbol{z})$ is improper, even $q_{\text{ood}}(\boldsymbol{z})$ for OOD data may be closer to $p(\boldsymbol{z})$ than $q_{\text{id}}(\boldsymbol{z})$.

180    **Brief summary.** Through analyzing *overestimation* scenarios from simple to complex, the answer
181    to the question at the beginning of this part could be: *the prior distribution* $p(\mathbf{z}) = \mathcal{N}(\boldsymbol{0}, \mathbf{I})$ *is an*
182    *improper choice for VAE when modeling a complex data distribution* $p(\boldsymbol{x})$, leading to an overestimated
183    $D_{\text{KL}}(q_{\text{id}}(\boldsymbol{z})||p(\boldsymbol{z}))$ and further raising the *overestimation* issue in unsupervised OOD detection.

## 184   4   Alleviating VAE's *overestimation* in Unsupervised OOD Detection

185    In this section, we develop the **"AVOID"** method to alleviate the influence of two aforementioned
186    factors in Section 3, including **i)** post-hoc prior and **ii)** dataset entropy calibration, both of which are
187    implemented in a simple way to inspire related work and can be further investigated for improvement.

### 188   4.1   Post-hoc Prior Method for Factor I

To provide a more insightful view to investigate the relationship between $q_{\text{id}}(\boldsymbol{z})$, $q_{\text{ood}}(\boldsymbol{z})$, and $p(\boldsymbol{z})$, we use t-SNE [37] to visualize them in Figure 4. The visualization reveals that $p(\boldsymbol{z})$ cannot distinguish between the latent variables sampled from $q_{\text{id}}(\boldsymbol{z})$ and $q_{\text{ood}}(\boldsymbol{z})$, while $q_{\text{id}}(\boldsymbol{z})$ is clearly distinguishable from $q_{\text{ood}}(\boldsymbol{z})$. Therefore, to alleviate *overestimation*, we can explicitly modify the prior distribution $p(\boldsymbol{z})$ in Eq. (8) to force it to be closer to $q_{\text{id}}(\boldsymbol{z})$ and far from $q_{\text{ood}}(\boldsymbol{z})$, *i.e.*, decreasing $D_{\text{KL}}(q_{\text{id}}(\boldsymbol{z})||p(\boldsymbol{z}))$ and increasing $D_{\text{KL}}(q_{\text{ood}}(\boldsymbol{z})||p(\boldsymbol{z}))$.

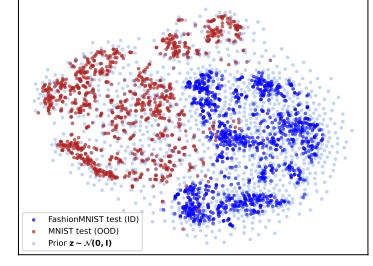

Figure 4: The t-SNE visualization of the latent representations on FashionM-NIST(ID)/MNIST(OOD) dataset pair.

A straightforward modifying approach is to replace $p(\boldsymbol{z})$ in ELBO with an additional distribution $\hat{q}_{\text{id}}(\boldsymbol{z})$ that can fit $q_{\text{id}}(\boldsymbol{z})$ well after training, where the target value of $q_{\text{id}}(\boldsymbol{z})$ can be acquired by marginalizing $q_\phi(\boldsymbol{z}|\boldsymbol{x})$ over the training set, *i.e.*, $q_{\text{id}}(\boldsymbol{z}) = \mathbb{E}_{\boldsymbol{x}\sim p_{\text{id}}(\boldsymbol{x})}[q_\phi(\boldsymbol{z}|\boldsymbol{x})]$. Previous study on distribution matching [30] has developed an LSTM-based method to efficiently fit $q_{\text{id}}(\boldsymbol{z})$ in the latent space, *i.e.*,

$$\hat{q}_{\text{id}}(\boldsymbol{z}) = \prod_{t=1}^{T} q(\boldsymbol{z}_t|\boldsymbol{z}_{<t}), \text{ where } q(\boldsymbol{z}_t|\boldsymbol{z}_{<t}) = \mathcal{N}(\mu_i, \sigma_i^2). \tag{12}$$

Thus, we could propose a "post-hoc prior" (PHP) method for Factor I, formulated as

$$\text{PHP}(\boldsymbol{x}) := \mathbb{E}_{\boldsymbol{z}\sim q_\phi(\boldsymbol{z}|\boldsymbol{x})} \log p_\theta(\boldsymbol{x}|\boldsymbol{z}) - D_{\text{KL}}(q_\phi(\boldsymbol{z}|\boldsymbol{x})||\hat{q}_{\text{id}}(\boldsymbol{z})), \tag{13}$$

which could lead to better OOD detection performance since it could enlarge the gap $\mathcal{G}$, *i.e.*,

$$\mathcal{G}_{\text{PHP}} = [-\mathcal{H}_{p_{\text{id}}}(\boldsymbol{x}) + \mathcal{H}_{p_{\text{ood}}}(\boldsymbol{x})] + [-D_{\text{KL}}(q_{\text{id}}(\boldsymbol{z})||\hat{q}_{\text{id}}(\boldsymbol{z})) + D_{\text{KL}}(q_{\text{ood}}(\boldsymbol{z})||\hat{q}_{\text{id}}(\boldsymbol{z}))] > \mathcal{G}. \tag{14}$$

Please note that PHP can be directly integrated into a trained VAE in a "plug-and-play" manner.

## 4.2 Dataset Entropy Calibration Method for Factor II

While the entropy of a dataset is a constant that remains unaffected by different model settings, it is still an essential factor that leads to *overestimation*. To address this, a straightforward approach is to design a calibration method that ensures the value added to the ELBO of ID data will be larger than that of OOD data. Specifically, we denote the calibration term as $\mathcal{C}(\boldsymbol{x})$, and its expected property could be formulated as

$$\mathbb{E}_{\boldsymbol{x}\sim p_{\text{id}}(\boldsymbol{x})}[\mathcal{C}(\boldsymbol{x})] > \mathbb{E}_{\boldsymbol{x}\sim p_{\text{ood}}(\boldsymbol{x})}[\mathcal{C}(\boldsymbol{x})]. \tag{15}$$

After adding the calibration $\mathcal{C}(\boldsymbol{x})$ to the ELBO$(\boldsymbol{x})$, we could obtain the "dataset entropy calibration" (DEC) method for Factor II, formulated as

$$\text{DEC}(\boldsymbol{x}) := \mathbb{E}_{\boldsymbol{z}\sim q_\phi(\boldsymbol{z}|\boldsymbol{x})} \log p_\theta(\boldsymbol{x}|\boldsymbol{z}) - D_{\text{KL}}(q_\phi(\boldsymbol{z}|\boldsymbol{x})||p(\boldsymbol{z})) + \mathcal{C}(\boldsymbol{x}). \tag{16}$$

With the property in Eq. (15), we could find that the new gap $\mathcal{G}_{\text{DEC}}$ becomes larger than the original gap $\mathcal{G}$ based solely on ELBO, as $\mathcal{G}_{\text{DEC}} = \mathcal{G} + \mathbb{E}_{\boldsymbol{x}\sim p_{\text{id}}(\boldsymbol{x})}[\mathcal{C}(\boldsymbol{x})] - \mathbb{E}_{\boldsymbol{x}\sim p_{\text{ood}}(\boldsymbol{x})}[\mathcal{C}(\boldsymbol{x})] > \mathcal{G}$, which should alleviate the *overestimation* and lead to better unsupervised OOD detection performance.

**How to design the calibration $\mathcal{C}(\boldsymbol{x})$?** For the choice of the function $\mathcal{C}(\boldsymbol{x})$, inspired by the previous work [13], we could use image compression methods like Singular Value Decomposition (SVD) [38] to roughly measure the complexity of an image, where the images from the same dataset should have similar complexity. An intuitive insight into this could be shown in Figure 5, where the ID dataset's statistical feature, *i.e.*, the curve, is distinguishable to other datasets. Based on this empirical study, we could first propose a **non-scaled** calibration function, denoted as $\mathcal{C}_{\text{non}}(\boldsymbol{x})$. First, we could set the number of singular values as $n_{\text{id}}$, which can achieve the reconstruction error $|\boldsymbol{x}_{\text{recon}} - \boldsymbol{x}| = \epsilon$ in the ID training set; then for a test input $\boldsymbol{x}_i$, we use SVD to calculate the smallest $n_i$ that could also achieve a smaller reconstruction error $\epsilon$, then $\mathcal{C}_{\text{non}}(\boldsymbol{x})$ could be formulated as:

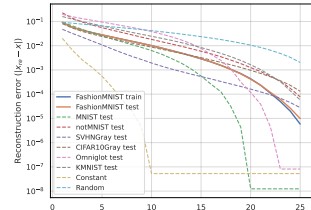

Figure 5: Visualization of the relationship between the number of singular values and the reconstruction error.

$$\mathcal{C}_{\text{non}}(\boldsymbol{x}) = \begin{cases} (n_i/n_{\text{id}}), & \text{if } n_i < n_{\text{id}}, \\ [((n_{\text{id}} - (n_i - n_{\text{id}}))/n_{\text{id}}], & \text{if } n_i \geq n_{\text{id}}, \end{cases} \tag{17}$$

which can give the ID dataset a higher expectation $\mathbb{E}_{\boldsymbol{x} \sim p_{\mathrm{id}}(\boldsymbol{x})}[\mathcal{C}_{\mathrm{non}}(\boldsymbol{x})]$ than that of other statistically different OOD datasets. More details to obtain $\mathcal{C}_{\mathrm{non}}(\boldsymbol{x})$ can be found in Appendix D.

## 4.3 Putting Them Together to Get "AVOID"

By combining the post-hoc prior (PHP) method and the dataset entropy calibration (DEC) method, we could develop a new score function, denoted as $\mathcal{S}_{\mathrm{AVOID}}(\boldsymbol{x})$:

$$\mathcal{S}_{\mathrm{AVOID}}(\boldsymbol{x}) := \mathbb{E}_{q_\phi(\boldsymbol{z}|\boldsymbol{x})}\left[\log p_\theta(\boldsymbol{x}|\boldsymbol{z})\right] - D_{\mathrm{KL}}(q_\phi(\boldsymbol{z}|\boldsymbol{x})||\hat{q}_{\mathrm{id}}(\boldsymbol{z})) + \mathcal{C}(\boldsymbol{x}). \tag{18}$$

To balance the importance of PHP and DEC terms in Eq. (18), we consider to set an appropriate scale for $\mathcal{C}(\boldsymbol{x})$. For the scale of $\mathcal{C}(\boldsymbol{x})$, if it is too small, its effectiveness in alleviating *overestimation* could be limited. Otherwise, it may hurt the effectiveness of the PHP method since DEC will dominate the value of "AVOID". Additionally, for statistically similar datasets, *i.e.*, $\mathcal{H}_{p_{\mathrm{id}}}(\boldsymbol{x}) \approx \mathcal{H}_{p_{\mathrm{ood}}}(\boldsymbol{x})$, the property in Eq. (15) cannot be guaranteed and we may only have $\mathbb{E}_{\boldsymbol{x} \sim p_{\mathrm{id}}(\boldsymbol{x})}[\mathcal{C}_{\mathrm{non}}(\boldsymbol{x})] \approx \mathbb{E}_{\boldsymbol{x} \sim p_{\mathrm{ood}}(\boldsymbol{x})}[\mathcal{C}_{\mathrm{non}}(\boldsymbol{x})]$, in which case we could only rely on the PHP method. Thus, an appropriate scale of $\mathbb{E}_{\boldsymbol{x} \sim p_{\mathrm{id}}(\boldsymbol{x})}[\mathcal{C}(\boldsymbol{x})]$, named "$\mathcal{C}_{\mathrm{scale}}$", could be derived by $\mathcal{C}_{\mathrm{scale}} = \mathbb{E}_{\boldsymbol{x} \sim p_{\mathrm{id}}(\boldsymbol{x})}[\mathrm{PHP}(\boldsymbol{x})] \approx \mathcal{H}_{p_{\mathrm{id}}}(\boldsymbol{x})$, which leads to

$$\mathbb{E}_{\boldsymbol{x} \sim p_{\mathrm{id}}(\boldsymbol{x})}[\mathrm{DEC}(\boldsymbol{x})] = -\mathcal{H}_{p_{\mathrm{id}}}(\boldsymbol{x}) - D_{\mathrm{KL}}(q_{\mathrm{id}}(\boldsymbol{z})||p(\boldsymbol{z})) + \mathcal{C}_{\mathrm{scale}} \approx -D_{\mathrm{KL}}(q_{\mathrm{id}}(\boldsymbol{z})||p(\boldsymbol{z})). \tag{19}$$

Thus, when $\mathcal{H}_{p_{\mathrm{id}}}(\boldsymbol{x}) \approx \mathcal{H}_{p_{\mathrm{ood}}}(\boldsymbol{x})$ and $\mathbb{E}_{\boldsymbol{x} \sim p_{\mathrm{id}}(\boldsymbol{x})}[\mathcal{C}(\boldsymbol{x})] \approx \mathbb{E}_{\boldsymbol{x} \sim p_{\mathrm{ood}}(\boldsymbol{x})}[\mathcal{C}(\boldsymbol{x})]$, the PHP part of "AVOID" could still be helpful to alleviate *overestimation*.

Motivated by the above analysis, we could implement the **scaled** calibration function, formulated as

$$\mathcal{C}(\boldsymbol{x}) = \mathcal{C}_{\mathrm{non}}(\boldsymbol{x}) \times \mathcal{C}_{\mathrm{scale}} = \begin{cases} (n_i/n_{\mathrm{id}}) \times \mathcal{C}_{\mathrm{scale}}, & \text{if} \quad n_i < n_{\mathrm{id}}, \\ [((n_{\mathrm{id}} - (n_i - n_{\mathrm{id}}))/n_{\mathrm{id}}] \times \mathcal{C}_{\mathrm{scale}}, & \text{if} \quad n_i \geq n_{\mathrm{id}}. \end{cases} \tag{20}$$

## 5 Experiments

### 5.1 Experimental Setup

**Datasets.** In accordance with existing literature [17, 18, 39], we evaluate our method against previous works using two standard dataset pairs: FashionMNIST [40] (ID) / MNIST [41] (OOD) and CIFAR10 [42] (ID) / SVHN [43] (OOD). The suffixes "ID" and "OOD" represent in-distribution and out-of-distribution datasets, respectively. To more comprehensively assess the generalization capabilities of these methods, we incorporate additional OOD datasets, the details of which are available in Appendix E.1. Notably, datasets featuring the suffix "-G" (e.g., "CIFAR10-G") have been converted to grayscale, resulting in a single-channel format.

**Evaluation and Metrics.** We adhere to the previous evaluation procedure [17, 18], where all methods are trained using the training split of the in-distribution dataset, and their OOD detection performance is assessed on both the testing split of the in-distribution dataset and the OOD dataset. In line with previous works [1, 5, 44], we employ evaluation metrics including the area under the receiver operating characteristic curve (AUROC ↑), the area under the precision-recall curve (AUPRC ↑), and the false positive rate at 80% true positive rate (FPR80 ↓). The arrows indicate the direction of improvement for each metric.

**Baselines.** Our experiments primarily encompass two comparison aspects: **i**) evaluating our novel score function "AVOID" against previous unsupervised OOD detection methods to determine whether it can achieve competitive performance; and **ii**) comparing "AVOID" with VAE's ELBO to assess whether our method can mitigate *overestimation* and yield improved performance. For comparisons in **i**, we can categorize the baselines into three groups, as outlined in [18]: "**Supervised**" includes supervised OOD detection methods that utilize in-distribution data labels [1, 5, 9, 45, 46, 47, 48, 49]; "**Auxiliary**" refers to methods that employ auxiliary knowledge gathered from OOD data [13, 39, 44]; and "**Unsupervised**" encompasses methods without reliance on labels or OOD-specific assumptions [14, 17, 18, 26]. For comparisons in **ii**, we compare our method with a standard VAE [25], which also serves as the foundation of our method. Further details regarding these baselines and their respective categories can be found in Appendix E.2.

**Implementation Details.** The VAE's latent variable $\boldsymbol{z}$'s dimension is set as 200 for all experiments with the encoder and decoder parameterized by a 3-layer convolutional neural network, respectively.

Table 1: The comparisons of our method and other OOD detection methods. The best results achieved by the methods of the category "Not ensembles" of "Unsupervised" have been bold.

| FashinMNIST(ID)/MNIST(OOD) | | | | | | CIFAR10(ID)/SVHN(OOD) | | | | | |
| Supervised | | Auxiliary | | Unsupervised | | Supervised | | Auxiliary | | Unsupervised | |
| Method | AUROC↑ | Mehod | AUROC↑ | Method | AUROC↑ | Method | AUROC↑ | Mehod | AUROC↑ | Method | AUROC↑ |
| CP [1] | 73.4 | LR(PC) [39] | 99.4 | *-Ensembles* | | MD [46] | 99.7 | LR(PC) [39] | 93.0 | *-Ensembles* | |
| CP(Ent) [1] | 74.6 | LR(BC) [39] | 45.5 | WAIC(5VAE) [26] | 76.6 | LMD [47] | 27.9 | LR(VAE) [39] | 26.5 | WAIC(5Glow) [26] | 99.0 |
| ODIN [45] | 75.2 | CP(OOD) [39] | 87.7 | WAIC(5PC) [26] | 22.1 | EN [6] | 98.9 | OE [44] | 98.4 | WAIC(5PC) [26] | 62.8 |
| VIB [5] | 94.1 | CP(Cal) [39] | 90.4 | *-Not Ensembles* | | iDE [52] | 95.7 | IC(Glow) [13] | 95.0 | *-Not Ensembles* | |
| MD(CNN) [46] | 94.2 | IC(Glow) [13] | 99.8 | LRe [14] | 98.8 | LN[9] | 98.4 | IC(PC++) [13] | 92.9 | LRe [14] | 87.5 |
| MD(DN) [46] | 98.6 | IC(PC++) [13] | 96.7 | HVK [17] | 98.4 | ODIN [45] | 82.9 | IC(HVAE) [13] | 83.3 | HVK [17] | 89.1 |
| DE [1] | 85.7 | | | $\mathcal{LLR}^{ada}$[18] | 98.0 | GN [49] | 76.7 | | | $\mathcal{LLR}^{ada}$[18] | 94.2 |
| | | | | AVOID(ours) | **99.2** | | | | | AVOID(ours) | **94.5** |

Table 2: The comparisons of our method with post-hoc prior (denoted as "PHP") or dataset entropy calibration (denoted as "DEC") individually and other unsupervised OOD detection methods. "PHP+DEC" is equal to our method "AVOID". Bold numbers are superior results.

| FashinMNIST(ID)/MNIST(OOD) | | | | CIFAR10(ID)/SVHN(OOD) | | | |
| Method | AUROC↑ | AUPRC↑ | FPR80↓ | Method | AUROC↑ | AUPRC↑ | FPR80↓ |
| ELBO [25] | 23.5 | 35.6 | 98.5 | ELBO [25] | 24.9 | 36.7 | 94.6 |
| WAIC(5PC) [26] | 22.1 | 40.1 | 91.1 | WAIC(5PC) [26] | 62.8 | 61.6 | 65.7 |
| HVK [17] | 98.4 | 98.4 | 1.3 | HVK [17] | 89.1 | 87.5 | 17.2 |
| $\mathcal{LLR}^{ada}$[18] | 97.0 | 97.6 | 0.9 | $\mathcal{LLR}^{ada}$[18] | 92.6 | 91.8 | 11.1 |
| *-Ours:* | | | | *-Ours:* | | | |
| PHP | 89.7 | 90.3 | 13.3 | PHP | 39.6 | 42.6 | 85.7 |
| DEC | 34.1 | 40.7 | 92.5 | DEC | 87.8 | 89.9 | 17.8 |
| PHP+DEC | **99.2** | **99.4** | **0.00** | PHP+DEC | **94.5** | **95.3** | **4.24** |

The reconstruction likelihood distribution is modeled by a discretized mixture of logistics [20]. For optimization, we adopt the same Adam optimizer [50] with a learning rate of 1e-3. We train all models in comparison by setting the batch size as 128 and the max epoch as 1000. All experiments are performed on a PC with an NVIDIA A100 GPU and our code is implemented with PyTorch [51]. More implementation details can be found in Appendix E.3.

## 5.2 Comparison with Unsupervised OOD Detection Baselines

First, we compare our method with other SOTA baselines in Table 1. The results demonstrate that our method achieves competitive performance compared to "Supervised" and "Auxiliary" methods and outperforms "Unsupervised" OOD detection methods. Next, we provide a more detailed comparison with some unsupervised methods, particularly the ELBO of VAE, as shown in Table 2. These results indicate that our method effectively mitigates *overestimation* and enhances OOD detection performance when using VAE as the backbone. Lastly, to assess our method's generalization capabilities, we test it on a broader range of datasets, as displayed in Table 3. Experimental results strongly verify our analysis of the VAE's *overestimation* issue and demonstrate that our method consistently mitigates *overestimation*, regardless of the type of OOD datasets.

## 5.3 Ablation Study on Verifying the Post-hoc Prior Method

To evaluate the effectiveness of the Post-hoc Prior (PHP), we compare it with other unsupervised methods in Table 2. Moreover, we test the PHP method on additional datasets and present the results in Table 4 of Appendix F. The experimental results demonstrate that the PHP method can alleviate the *overestimation*. To provide a better understanding, we also visualize the density plot of ELBO and PHP for the "FashionMNIST(ID)/MNIST(OOD)" dataset pair in Figures 6(a) and 6(b), respectively.

The Log-likelihood Ratio ($\mathcal{LLR}$) methods [17, 18] are the current SOTA unsupervised OOD detection methods that also focus on latent variables. These methods are based on an empirical assumption that the bottom layer latent variables of a hierarchical VAE could learn low-level features and top layers learn semantic features. However, we discovered that while ELBO could already perform well in detecting some OOD data, the $\mathcal{LLR}$ method [18] could negatively impact OOD detection performance to some extent, as demonstrated in Figure 6(c), where the model is trained on MNIST and detects FashionMNIST as OOD. On the other hand, our method can still maintain comparable performance since the PHP method can explicitly alleviate *overestimation*, which is one of the strengths of our method compared to the SOTA methods.

## 5.4 Ablation Study on Verifying the Dataset Entropy Calibration Method

We evaluate the performance of dataset entropy calibration, referred to as "DEC", in Table 2 and Table 5 of Appendix G. Although the DEC method is simple, our results show that it effectively alleviates *overestimation*. To better understand DEC, we visualize the calculated $\mathcal{C}(\boldsymbol{x})$ of CIFAR10

Table 3: The comparisons of our method "AVOID" and baseline "ELBO" on more datasets. Bold numbers are superior performance.

| ID | FashionMNIST | | | ID | CIFAR10 | | |
|---|---|---|---|---|---|---|---|
| OOD | AUROC ↑ | AUPRC ↑ | FPR80 ↓ | OOD | AUROC ↑ | AUPRC ↑ | PFR80 ↓ |
| | ELBO / AVOID (ours) | | | | ELBO / AVOID (ours) | | |
| KMNIST | 60.03 / **78.71** | 54.60 / **68.91** | 61.6 / **48.4** | CIFAR100 | 52.91 / **55.36** | 51.15 / **72.13** | 77.42 / **73.93** |
| Omniglot | 99.86 / **100.0** | 99.89 / **100.0** | 0.00 / **0.00** | CelebA | 57.27 / **71.23** | 54.51 / **72.13** | 69.03 / **54.45** |
| notMNIST | 94.12 / **97.72** | 94.09 / **97.70** | 8.29 / **2.20** | Places365 | 57.24 / **68.37** | 56.96 / **69.05** | 73.13 / **62.64** |
| CIFAR10-G | 98.01 / **99.01** | 98.24 / **99.04** | 1.20 / **0.40** | LFWPeople | 64.15 / **67.72** | 59.71 / **68.81** | 59.44 / **54.45** |
| CIFAR100-G | 98.49 / **98.59** | 97.49 / **97.87** | 1.00 / **1.00** | SUN | 53.14 / **63.09** | 54.48 / **63.32** | 79.52 / **68.63** |
| SVHN-G | 95.61 / **96.20** | 96.20 / **97.41** | 3.00 / **0.40** | STL10 | 49.37 / **64.51** | 47.79 / **65.50** | 78.02 / **67.23** |
| CelebA-G | 97.33 / **97.87** | 94.71 / **95.82** | 3.00 / **0.40** | Flowers102 | 67.68 / **76.83** | 64.68 / **78.01** | 57.94 / **46.65** |
| SUN-G | 99.16 / **99.32** | 99.39 / **99.47** | 0.00 / **0.00** | GTSRB | 39.50 / **53.06** | 41.73 / **49.84** | 86.61 / **73.63** |
| Places365-G | 98.92 / **98.89** | 98.05 / **98.61** | 0.80 / **0.80** | DTD | 37.86 / **81.82** | 40.93 / **62.42** | 82.22 / **64.24** |
| Const | 94.94 / **95.20** | 97.27 / **97.32** | 1.80 / **1.70** | Const | 0.001 / **80.12** | 30.71 / **89.42** | 100.0 / **22.38** |
| Random | 99.80 / **100.0** | 99.90 / **100.0** | 0.00 / **0.00** | Random | 71.81 / **99.31** | 82.89 / **99.59** | 85.71 / **0.000** |

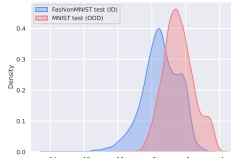 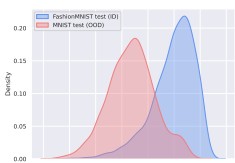 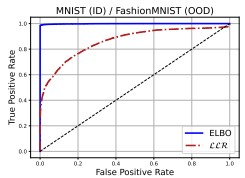 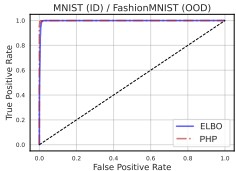

(a) Density plot of ELBO    (b) Density plot of PHP    (c) ROC curve of $\mathcal{LLR}$    (d) ROC curve of PHP

Figure 6: Density plots and ROC curves. **(a):** directly using ELBO($\boldsymbol{x}$), an estimation of the $p(\boldsymbol{x})$, of a VAE trained on FashionMNIST leads to *overestimation* in detecting MNIST as OOD data; **(b):** using PHP method could alleviate the *overestimation*; **(c):** SOTA method $\mathcal{LLR}$ hurts the performance when ELBO could already work well; **(d):** PHP method would not hurt the performance.

(ID) in Figure 7(a) and other OOD datasets in Figure 7(b) when $n_{id} = 20$. Our results show that the $\mathcal{C}(\boldsymbol{x})$ of CIFAR10 (ID) achieves generally higher values than that of other datasets, which is the underlying reason for its effectiveness in alleviating *overestimation*. Additionally, we investigate the impact of different $n_{id}$ on OOD detection performance in Figure 7(c), where our results show that the performance is consistently better than ELBO.

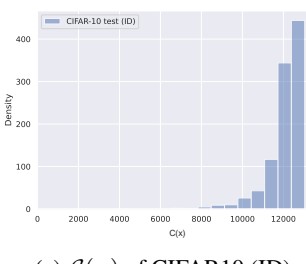 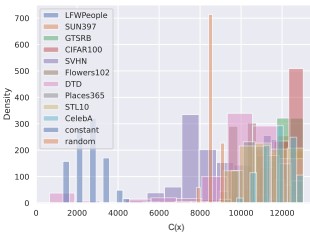 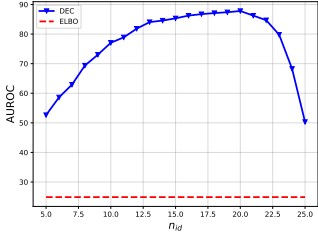

(a) $\mathcal{C}(\boldsymbol{x})$ of CIFAR10 (ID)    (b) $\mathcal{C}(\boldsymbol{x})$ of OOD datasets    (c) Impact of $n_{id}$

Figure 7: **(a)** and **(b)** are respectively the visualizations of the calculated entropy calibration $\mathcal{C}(\boldsymbol{x})$ of CIFAR10 (ID) and other OOD datasets, where the $\mathcal{C}(\boldsymbol{x})$ of CIFAR10 (ID) could achieve generally higher values. **(c)** is the OOD detection performance of dataset entropy calibration with different $n_{id}$ settings, which consistently outperforms ELBO.

## 6 Conclusion

In conclusion, we have identified the underlying factors that lead to VAE's *overestimation* in unsupervised OOD detection: the improper design of the prior and the gap of the dataset entropies between the ID and OOD datasets. With this analysis, we have developed a novel score function called "AVOID", which is effective in alleviating *overestimation* and improving unsupervised OOD detection. This work may lead a research stream for improving unsupervised OOD detection by developing more efficient and sophisticated methods aimed at optimizing these revealed factors.

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
