# OpenReview forum: "Enemy is Inside: Alleviating VAE's Overestimation in Unsupervised OOD Detection"
_NeurIPS.cc/2023/Conference — Submitted to NeurIPS 2023_

### Official Review · Reviewer_GLsZ · 2023-06-29

**Soundness:** 2 fair
**Presentation:** 3 good
**Contribution:** 2 fair
**Rating:** 5
**Confidence:** 3

**Summary:**

The paper first mathematically examines the unsupervised (without training label) OOD detection performance using VAE, decomposing the expected ELBO into two components: (i) entropy $\mathcal{H}(x)$ of a dataset, (ii) KL divergence $D_{KL}(q(z)||p(z))$ between the estimated $z$ and the prior. It's theoretically shown that the entropy of data distribution is defined by itself, thus may not bring benefit to the OOD detection problem (Eq. 8). Then the paper mathematical and empirically analyzes the second component. The paper shows in some simple cases the prior $p(z)$ and the dataset $p(x)$ can not fit well with the VAE model, which results in for some $x$, $p_\theta(x)$ estimated by the trained VAE model has high value when $p(x)$ has low value, which is the overestimation problem for OOD detection. The paper proposes to use post-hoc prior method (estimate the prior from the trained VAE and the ID dataset) to revise the issue of the improper design of prior and add calibration to alleviate the issue of entropy. Empirical results show that the proposed AVOID method constantly improves the OOD detection performance simply based on ELBO (Table 3), and outperforms existing unsupervised non-ensemble OOD detection methods.

**Strengths:**

1. The paper mathematically examines the unsupervised (without training label) OOD detection performance using VAE, decomposing the expected ELBO into two components: (i) entropy $\mathcal{H}(x)$ of a dataset, (ii) KL divergence $D_{KL}(q(z)||p(z))$ between the estimated $p(z)$ and the prior, which is quite crucial to understand the underline benefit and drawback to use ELBO as OOD score.
2. The demos including Figure 2,3,4 shows the improper of the traditionally chosen prior leading to the mismatch between prior and post-hoc prior, and the high probability of OOD sample in the prior. The observation well inspires the proposed post-hoc prior method.
3. Experiment includes varied OOD detection methods including supervised, auxiliary, and unsupervised (ensemble/non-ensemble), and shows the proposed method beats baselines within a specific category.


**Weaknesses:**

1. Notation is not consistent such as $p$ and $p_\theta$ in Figure 3.
2. Eq. 8 uses the entropy difference between ID and OOD distributions. Eq. 8 tells us the more diverse the ID distribution, the harder the OOD detection task. I think the OOD here should consider overall OOD distribution instead of an OOD dataset distribution. If not, I can simply define each OOD data point as a distribution which has $\mathcal{H}_{p_o}=1$ or I consider overall OOD data together (overall OOD distribution) which may have a pretty large diversity and very low entropy. Thus the motivation for the second method is not well held. I believe the idea of the second method is good itself, it leverages some extra information to improve the OOD performance.
3. Sec. 3.1 uses 3-layer NN for $q_\phi$ and $p_\theta$. The dataset is synthetic, thus I wonder whether increasing the number of training samples and NN capability would help better estimate $p_\theta(x)$. In other words, the reason that ELBO suffers from overestimation is the number of training samples, NN capability, or something else. Or perhaps the observation from Figure 3 is even when ID is well estimated, the OOD is still not well estimated.


**Questions:**

(1) I would like to ask for thoughts on Eq. (8), what's the right definition for $\mathcal{H}_{p_o}$? I feel the choices include: (i) regards only 1 sample as $p_o$, (ii) regards some OOD samples as $p_o$, (iii) regards overall OOD samples as $p_o$. I think (iii) is the best choice though it's not feasible, and (i) and (ii) are not the correct choice. The DEC itself is an OOD detection method and can be ensembled into any OOD detection method, thus I feel it's less related to factor two, the entropy issue.
(2) Why in Table 2 PHP performs better than DEC on the left table and perform worse on the right table?

---

> ### Author Rebuttal · Authors · 2023-08-10
>
> **For weakness 1:** Thanks for your careful reading and we will correct the inconsistent notation.
>
> **For weakness 2 & question 1:**
> We pretty much appreciate this thoughtful comment. In the beginning, please allow us to point out a factual error in your comments that the entropy of a dataset will become larger with the increase of the diversity (not "large diversity and very low entropy''), but it does not matter us to understand the high-level idea of your comments.
>
> For the definition of the data distribution $p_{id}(x)$ and $p_{ood}(x)$, it is reasonable to define the $p_{id}(x)$ as an underlying distribution where the observed training set $\\{x_i\\}^N_{i=1}$ is sampled i.i.d. N times from, but the definition of $p_{ood}(x)$ could be empirical and diverse. From the perspective of DEC method, given a testing sample $x_{ood}^j$, its corresponding $p_{ood}$ should be defined as a distribution where most samples with higher probabilities for generation from it are **semantically** (like category) and **statistically** (like image complexity) similar to $x_{ood}^j$. Then, it requires us to estimate  $\mathcal{H}\_{ood}$ defined on $p_{ood}$ with the given testing sample $x_{ood}$.
> If we have sufficient OOD data points sampled from $x_{ood}^j$'s corresponding $p_{ood}$, we can provide a more precise estimation for $\mathcal{H}\_{p_{ood}}$ though it is not feasible in practice.
>
>
> We also agree with your point that the ground truth of $\mathcal{H}\_{p_{ood}}$ would be a const, no matter 1 for uniform distribution or 0 for impulse distribution, but you can never know the amount of OOD samples during testing, whose challenge relies on requiring us to use only one OOD data sample to estimate $\mathcal{H}\_{p_{ood}}$ on the whole OOD dataset.
>
>
> Although it is hard to get the ground truth $\mathcal{H}\_{p_{ood}}(x)$, we provide an approximating method for it with DEC method. There are two intuitions:
> 1) the images in a dataset have similar complexity (complexity is related to the low-level feature like texture, e.g., CIFAR-10 has more complexity than MNIST), where the complexity could be evaluated by image compression methods like SVD, i.e., a more complex image needs more bits (like more singular values) to compress in order to reconstruct the image to a certain degree of the reconstruction error. Figure 7 could empirically support this intuition;
> 2) a dataset containing images with higher image complexity should have higher entropy than the simpler ones, e.g., CIFAR-10 (categories including cat, dog, horse, etc.) is much more complex than the SVHN dataset (House numbers), which indicates the underlying $p_{cifar}(x)$ is much more diverse than $p_{svhn}(x)$ and leads to higher entropy.
>
> Finally, the DEC method scales the image complexity $\mathcal{C}_{non}(x)$ to $\mathcal{C}(x)$, which could have a similar scale of the entropy $\mathcal{H}(x)$.
>
>
> **For weakness 3:** Increasing the model capacity has been proved to be useless in previous works [17, 18], where the overestimation issue still exists. Considering the number of training samples could be interesting, thus we add additional experiments in **Table 3 and Figure 1 \(a-b\) of the one-page rebuttal pdf** to evaluate the influence of the number of training samples and model capacity (number of nn layers) in the synthetic and practical datasets. The conclusion is that influence of number of the training samples in alleviating overestimation is very limited and increasing the model capacity could not bring significant improvement.
>
>
> **For question 2:** The performance of PHP and DEC in improving OOD detection is independent, since they are focusing on totally different factors. When the $q_{id}(z)$ is very distinguishable to $q_{ood}(z)$, PHP method could perform better. When the dataset entropy (or image complexity) is extremely huge between ID and OOD datasets, the DEC method could perform better.

---

> > ### Comment · Reviewer_GLsZ · 2023-08-10
> > **Response to rebuttal.**
> >
> > Thank you for your rebuttal!
> >
> > Sorry, it's my mistake, a dataset with high diversity corresponds to high entropy.
> >
> > Thank you for your further explanation. How complexity and entropy got connected is still confusing me. Let me discuss by the definitions of the terms:
> >
> > (a) entropy of $x$ of mnist dataset $x \sim D_x^{mnist}$, $p^{mnist}(x)$: $\int -p^{mnist}(x) log(p^{mnist}(x)) dx$
> >
> > (b) the density of $x$ in the overall OOD dataset $x \sim D_x^{overall}$: $p^{overall}(x)$
> >
> > (c) complexity of $x$: approximate via image compress method.
> >
> > Then let me share what I think about the connection between (a), (b), and (c):
> >
> > (1)  "given a testing sample $x_{ood}^j$, its corresponding $p_{ood}$ should be defined as a distribution $D_x$ where most samples with higher probabilities for generation from it are semantically (like category) and statistically (like image complexity) similar to $x_j$". It's a vague definition and It's weird to define $x$ first and then define the $D_x$. We should have $D_x$ first, then $x_{ood}^j \in D_x$ could be on the boundary of $D_x$.
> >
> > (2) "a dataset containing images with higher image complexity should have higher entropy than the simpler ones." It also depends on how many classes are in the dataset. I can combine two different cifar10 to make a dataset cifar20 with bigger entropy than cifar10, ie, images in cifar20 have similar complexity as cifar10, but cifar20 has a higher entropy. I believe the higher complexity implies a lower density (I'm connecting (c) and (b)). The quoted sentence in (2) is not aligned with (1), (2) is discussing a dataset while (1) is for one sample. I believe this is caused by some notation/terminology confusion.
> >
> > (3) If (c) is connected to (b) via higher complexity of $x$ implying a lower $p^{overall}(x)$, (this is actually related to the quoted sentence in (1)), then how is it connected with (a), a dataset entropy in equation (8)?

---

> > > ### Author Response · Authors · 2023-08-14
> > > **The DEC method is NOT estimating dataset entropy by image complexity**
> > >
> > > Thank you for affording us a second opportunity to engage in further discussion about our paper!
> > > Firstly, we apologize for not recognizing the significant misunderstanding that may have arisen due to the unclear explanations in our initial response. We intend to address this confusion and start with your comments regarding the relationship between image complexity and dataset entropy.
> > >
> > > * (1) "It's a vague definition ..."
> > >
> > >     Thanks for your correction, we agree with your point that we should have $D_x$ first, and then we can define a data sample $x^j \in D_x$. Let's make it clearer based on your definitions.
> > >
> > >     i) an ID dataset $D_{id}$ is sampled from $p_{id}(x)$, and each ID sample satisfies $x_{id}^j \in D_{id}$;
> > >
> > >     ii) an OOD dataset $D_{ood}$ is sampled from $p_{ood}(x)$, and each OOD sample satisfies $x_{ood}^j \in D_{ood}$;
> > >
> > >     We hope to reach an agreement on the definition of $D_{ood}$. As stated in your first comment, we agree with the point that you can define $D_{ood}$ as no matter a single data sample, a set of limited data samples, or unlimited data samples. The ultimate use of these datasets $D_{ood}$ with various data sizes is still the same, which is to be used to estimate some metrics conditioned on $p_{ood}(x)$, like $\mathbb{E}\_{p}[ELBO(x)]\approx\mathbb{E}\_{D_x}[ELBO(x)]$. Moreover, it doesn't affect the analysis in our paper that a higher $\mathcal{H}\_{p_{id}}$ and $D_{KL}[q_{id}(z)||p(z)]$ would cause the gap $\mathcal{G}$ in Eq. (8) to be smaller, i.e., easier for an overestimation issue to occur.
> > >
> > > * (2) "I believe the higher complexity implies a lower density..."
> > >
> > >     (3) "how is it connected with (a)..."
> > >
> > >     Thanks for your vivid example ("cifar20") to demonstrate the relationship between image complexity and dataset entropy. Actually, we are still not sure if the conclusion "the higher complexity implies a lower density" is correct, because the image complexity in your example won't change with the dataset entropy. Thus, we tend to believe there is no direct relationship between a single image's complexity and a dataset's entropy.
> > >
> > >
> > > [Below is also a response to **Reviewer LEk9**]
> > >
> > > However, no matter what the relationship between image complexity and dataset entropy is, it won't affect the effectiveness of our DEC method. Because **we don't use image complexity to estimate dataset entropy** in the original paper (Sorry for our last non-rigorous response).
> > >
> > >  The core idea of DEC is to add a calibration term $\mathcal{C}(x)$ to $ELBO(x)$, where $\mathcal{C}(x)$ should satisfy two properties:
> > >
> > > 1) $\mathbb{E}\_{p_{id}}[\mathcal{C}(x)] > \mathbb{E}\_{p_{ood}}[\mathcal{C}(x)]$ to make $\mathcal{G}$ in Eq. (8) to be larger to alleviate the overestimation of ELBO-based OOD detection methods (**no dataset entropy needs to be estimated here**);
> > > 2) $\mathbb{E}\_{p_{id}}[\mathcal{C}(x)]$ should have a similar scale of $\mathcal{H}\_{p_{id}}(x)$ to ensure the effectiveness of AVOID method as analyzed in line 237~246 (**$\mathcal{H}\_{p_{id}}(x)$ is estimated by $\mathcal{H}\_{p_{id}}(x) \approx \mathbb{E}\_{x\sim p_{id}}[PHP(x)]$ instead of image complexity**).
> > >
> > >
> > >
> > > For property 1, what we need is an effective method that could roughly discriminate the ID/OOD data (An ideal choice is a score function that assigns ID data "1" and OOD data "0"). Since OOD data samples may occur with semantic and/or statistical differences from ID ones and our previous PHP method has already focused on the semantic aspect, we hope the property 1 of the DEC method can be achieved from the perspective of **statistical difference**, which naturally leads to the score functions based on sample-level statistics.  Image complexity is one of the choices and it has been proven to be effective by our experiments.
> > >     As the definition of $\mathcal{C}(x)$ shown in Eq. (20) of our paper, $\mathbb{E}\_{p_{id}}[\mathcal{C}(x)] > \mathbb{E}\_{p_{ood}}[\mathcal{C}(x)]$ will constantly satisfy as long as the image complexity of OOD and ID samples is different.
> > >
> > >
> > >
> > >
> > >
> > >
> > > For property 2, we develop a scaled version of $\mathcal{C}(x)$ by introducing a task-adaptive scale factor $\mathcal{H}\_{p_{id}}(x)$, which is **NOT** estimated by image complexity but by $\mathcal{H}\_{p_{id}}(x)\approx \mathbb{E}\_{x\sim p_{id}}[PHP(x)]$.
> > >
> > >
> > > We admit that the DEC method based on SVD is not that perfect, because when the OOD data image complexity is similar to ID data, the SVD-based DEC method could "fail" (the contribution to alleviating overestimation is limited but it would **not** cause any harm) though AVOID method could still rely on the PHP method to alleviate overestimation.
> > >
> > > However, we note that the most important contribution of this work is firstly identifying two factors that cause the overestimation issue of VAE, and we also believe that there will be other more effective score functions to be discovered based on our findings, which can further improve the performance.

---

> > > ### Author Response · Authors · 2023-08-18
> > >
> > > Dear reviewer GLsZ,
> > >
> > > Thanks again for engaging in further discussion with us. We sincerely hope that our response has helped address your confusion regarding image complexity and dataset entropy. After clearing up the misunderstanding, please allow us to kindly request your valuable time to review the mechanism of our method and rejudge its contribution to the field of unsupervised OOD detection. Your thoughtful evaluation would be greatly appreciated.
> > >
> > > Best regards,
> > >
> > > Authors

---

> > > > ### Comment · Reviewer_GLsZ · 2023-08-19
> > > > **Response to Rebuttal**
> > > >
> > > > Factors 1 and 2 look good to me.
> > > > I'm also good with the proposed two methods working well. DEC looks good to me as a method.
> > > > What looks missing to me is how we come to DEC from Factor 2.
> > > > "However, no matter what the relationship between image complexity and dataset entropy is, it won't affect the effectiveness of our DEC method. Because we don't use image complexity to estimate dataset entropy in the original paper."
> > > > I wish to see in the paper that we show the connection between image complexity to dataset entropy, therefore we use complexity for Factor 2.
> > > > I still have concerns about the connection between image complexity to dataset entropy.
> > > > I will keep my score.

---

> > > > > ### Author Response · Authors · 2023-08-20
> > > > > **Further response to the remaining confusion**
> > > > >
> > > > > Thanks for your further discussion and suggestion!
> > > > > Firstly, we are glad that we have achieved an agreement on factor 2's contribution to the overestimation, specifically leading the value of $\mathcal{G}$ in Eq. (8) to be too small and even negative.
> > > > > Then, with the analysis of factor 2, the DEC method is proposed to satisfy the two properties, $\mathbb{E}\_{id}[\mathcal{C}(x_{id})]>\mathbb{E}\_{ood}[\mathcal{C}(x_{ood})]$ and $\mathbb{E}\_{id}[\mathcal{C}(x_{id})]\approx\mathcal{H}\_{id}$ as discussed in our secondary response and main paper, contributing to offset the negative impact caused by the gap of dataset entropies.
> > > > > Note that the image-complexity-based DEC method is only one of the effective ways to achieve the ultimate goal of the DEC method and it should also satisfy the two properties.
> > > > >
> > > > > The key point to understand the connection between the image-complexity-based DEC method to dataset entropy should be more on the property 2 ($\mathbb{E}\_{id}[\mathcal{C}(x_{id})]\approx\mathcal{H}\_{id}$), i.e. the scale of $\mathcal{C}(x_{id})$.
> > > > > Since it may be not that easy to follow the short explanation of the scale in the secondary response, we want to add a representative example here to demonstrate this key point:
> > > > >
> > > > > Consider a hard ID/OOD case that we need to detect the vertically flipped (VFlip) in-distribution testing sample as OOD for the in-distribution testing sample itself (ID), the image-complexity-based DEC method could be totally useless since ID and OOD data sample's complexity is exactly the same. Thus, in this case, we could only rely on the PHP method to do OOD detection (we also add experiments as suggested by reviewer ogin in https://openreview.net/forum?id=31zVEkOGYU&noteId=unI5jvIInm).
> > > > >
> > > > > However, if the scale of $\mathbb{E}\_{id}[\mathcal{C}(x_{id})]$ is too big, it will dominate the AVOID method and the PHP method could be useless in the above case; if the scale is too small, it cannot be enough to alleviate the overestimation induced by factor 2 in other cases, where the AVOID method could also suffer. Thus, the only proper scale is to let $\mathbb{E}\_{id}[\mathcal{C}(x_{id})]\approx\mathcal{H}\_{id}$, where $\mathcal{H}\_{id}$ is estimated by PHP method. With this scale, the detection for this **VFlip** case would be
> > > > > $$\mathbb{E}\_{id}[DEC(x_{id})] = -\mathcal{H}\_{id}-D_{KL}[q_{id}(z)||p(z)]+ \mathbb{E}\_{id}[\mathcal{C}(x_{id})] \approx -D_{KL}[q_{id}(z)||p(z)]$$ and
> > > > > $$\mathbb{E}\_{ood}[DEC(x_{ood})] = -\mathcal{H}\_{ood}-D_{KL}[q_{ood}(z)||p(z)]+ \mathbb{E}\_{ood}[\mathcal{C}(x_{ood})] \approx -D_{KL}[q_{ood}(z)||p(z)].$$
> > > > >
> > > > > Thus, the PHP method would not be affected by the DEC method, and AVOID method could still effectively detect the VFlip data samples as OOD, which has also been evidenced by the additional experiments.
> > > > >
> > > > >
> > > > > We hope this response, with a specific VFlip example case, could address your confusion, and we would greatly appreciate it if there are more suggestions!

---

> > > > > > ### Comment · Reviewer_GLsZ · 2023-08-22
> > > > > >
> > > > > > "$E[C(x_{id})]>E[C(x_{ood})]$'': it's a general request for any OOD detection method, thus we can not say we derived it from Factor 2.
> > > > > >
> > > > > > "the only proper scale is to let $E[C(x_{id})]\approx H_{id}$, where $H_{id}$ is estimated by PHP method.'' Thank you very much for your further explanation. I agree it's an explanation. But when I go back to see the method, we are actually ensembling two OOD detection methods with a weighted sum, we are ensembling them. How to choose weight? We get inspiration from $H_{in}$ and $H_{ood}$. I have not specified the method for the method until now. Does this mean we can choose an arbitrary OOD method as $C$? Why we should choose complexity? Does complexity represent the $H_{in}$ well? I cannot say yes. Because I think if I'm ensembling two different OOD methods, then it should work with good weight, there is no need for one of them to be the complexity-based DEC method.

---

> > > > > > > ### Author Response · Authors · 2023-08-22
> > > > > > > **Thank you very much for your further response!**
> > > > > > >
> > > > > > > Thank you very much for your further response and we are glad that we have achieved an agreement on the connection between factor 2 (dataset entropy) and the DEC method, i.e., "$\mathbb{E}\_{id}[\mathcal{C}(x_{id})]\approx\mathcal{H}\_{id}$".
> > > > > > > Yes, in our understanding, for ELBO-based OOD detection, we can choose an arbitrary score function as an implementation of the DEC term to detect OOD samples as long as it satisfies $\mathbb{E}\_{id}[\mathcal{C}(x_{id})]>\mathbb{E}\_{ood}[\mathcal{C}(x_{ood})]$, and the main difficulty is the choice of the scale of $\mathbb{E}\_{id}[\mathcal{C}(x_{id})]$, which can be various for different implementations of $\mathcal{C}(x)$ and a proper scale is required to make the final OOD detection method "AVOID" effective, a.k.a choose a "weight" for it.
> > > > > > >
> > > > > > >
> > > > > > > In the DEC method, we set this proper scale ("weight") as $\mathbb{E}\_{id}[\mathcal{C}(x_{id})]\approx \mathcal{H}\_{id}$ to make AVOID (you could call it as an "ensemble of two OOD detection methods") be effective in most cases, which has been demonstrated in the previous analysis with the "VFlip" example under the suggestion of reviewer ogin. We note that the usage of image complexity is only used to make sure the DEC method satisfies the condition $\mathbb{E}\_{id}[\mathcal{C}(x_{id})]>\mathbb{E}\_{ood}[\mathcal{C}(x_{ood})]$, and won't be used to estimate the scale $\mathcal{H}\_{id}$, which is estimated by the PHP method $\mathcal{H}\_{id}\approx \mathbb{E}\_{id}[PHP(x_{id})]$. And we believe that there could be other more efficient score functions to replace image complexity to make our DEC method more effective.
> > > > > > >
> > > > > > > Thus, we highly agree with you that "there is no need for one of them to be the complexity-based DEC method." and we have already discussed the reason for choosing image-complexity-based DEC in our secondary response:
> > > > > > > >"Since OOD data samples may occur with semantic and/or statistical differences from ID ones and our previous PHP method has already focused on the semantic aspect, we hope the property 1 of the DEC method can be achieved from the perspective of statistical difference, which naturally leads to the score functions based on sample-level statistics. Image complexity is **one of the choices** and it has been proven to be effective by our experiments."
> > > > > > >
> > > > > > > At last, we want to emphasize that the focus of this paper is to identify the two factors contributing to the overestimation of the ELBO-based OOD detection, and the developed PHP, DEC, and AVOID methods are designed for supporting the analysis. We believe the performance of AVOID could be further improved with the introduction of better methods focusing on the two factors.

---

### Official Review · Reviewer_LEk9 · 2023-07-02

**Soundness:** 2 fair
**Presentation:** 2 fair
**Contribution:** 3 good
**Rating:** 4
**Confidence:** 4

**Summary:**

The paper discusses the phenomenon of overestimation, the allocation of higher likelihoods to out-of-distribution data points, in deep generative models. It analyses two factors which may cause the overestimation problem specific to VAEs from a reformulation of the ELBO. These two factors are posterior collapse and a difference in entropies between in-distribution and out-of-distribution datasets. The paper proposes, again specific to VAEs, a method called AVOID for alleviating these two factors.

**Strengths:**

* The clear definition of overestimation in Eq. (3) following is useful, also for the wider literature.
* The experimental setup is large: Table 3 demonstrates that the number of dataset combinations considered is numerous, in particular in comparison to previous work.


**Weaknesses:**

* In 3.2, the authors pose the questions “When is the design of the prior proper/not proper”, but answer these questions by providing an example for each case. While this is useful for illustrative purposes, it does not answer the stated question. The first few examples are furthermore focussing on linear VAEs which are not relevant for common practical uses, which limits the relevance of the theoretical results in this section.
* The design of the calibration term in ll. 219 is unclear. In my opinion, it is not properly explained, and important choices like SVD are not well motivated. When would SVD likely fail? Why does SVD intuitively capture the difference in entropy between the datasets?  The words “complexity” and “entropy” seem to be used interchangeably, please explain or use consistently.
* The experimental results are difficult to interpret, it is partly not possible to draw meaningful insights from it. It is worth noting that this is common in similar works on alleviating OOD detection in DGMs, the methods are hard to compare due to different experimental setups. However, in this work, important questions I have are: 1) In Table 1, in the unsupervised column, why is AVOID highlighted in bold, even though WAIC  outperforms it sometimes? 2) Where is the performance of a standard VAE without any adaptations listed?  I find this an important benchmark. 3) What is the decision criterion for OOD vs. in-distribution? Is it a threshold on the amended likelihood? If yes, looking at the density plots of Fig. 6 (b), how is it possible that there is still  a lot of overlap between the two datasets in PHP, even though the accuracy according to Table 1 is 99.2%? This seems inconsistent to me. 4) The experimental results report no standard deviations in key tables, such as Table 1 and 2 . DGM based methods are well known to be unstable, hence standard deviations would be useful. However, Table 3 partly alleviates this problem due to the large number of dataset combinations considered. 5) Table 3: I would argue that comparing CIFAR10 and CIFAR100 (and possibly other combinations) seems meaningless: The datasets are overlapping, hence it is unclear what is OOD and what is in-distribution.
* The language is sometimes unclear, in general slightly hard to understand, and could be greatly improved.

In summary, while this work demonstrates a large effort and a clear analytical approach to alleviating the overestimation problem in VAEs, important questions remain unclear. I am open to reconsidering my score upon a response from the authors.


**Questions:**

* ll. 191: “The visualization reveals that p(z) cannot distinguish between the latent variables sampled from qid(z) and qood(z), while qid(z) is 193 clearly distinguishable from qood(z).” I don’t understand this sentence, could you please explain it? What do you mean by “the prior cannot distinguish latent samples”?
* In Eq. (16), the proposal seems to be to add a regularizer which accounts for calibration of differing entropies to the ELBO. Why is this a good choice? Should it be weighted with the remainder of the objective?
* The proposal for alleviating factor 1 is learning a more complicated, LSTM-based prior. How would a VAE which is end-to-end trained with this more complicated prior perform? Would any VAE with a more complicated prior (of which there are plenty) help alleviate the overestimation problem?
* The authors could consider discussing and benchmarking against this recent work, which provides an orthogonal, score-based approach to the problem of anomaly detection in DGMs: https://openreview.net/forum?id=deYF9kVmIX



**Limitations:**

* The overestimation problem is common to many DGM methods, but this work provides a solution for VAEs only. The scope is bigger, and one could argue that it might be more interesting to find the underlying root cause in all DGM methods which suffer from this problem (if there is one). Yet, considering VAEs is a very interesting start.

---

> ### Author Rebuttal · Authors · 2023-08-10
>
> **For weakness 1:**
> Thanks for this helpful suggestion. As shown in Lines 180~183 in our paper, we have summarized the answer to the stated question after enumerating several typical cases. Then we highlight that the analyzed case in Fig.3 is targeted at non-linear VAEs (note the activation function), and the conclusion can be flexibly extended for other non-linear VAEs with various network structures, such as the convolutional VAEs used in our experiments.
>
> To make the analysis more relevant to the common practical cases, we add an additional related experiment in **Table 3 and Figure 1 \(a & b\) of the "one-page rebuttal pdf"**. We testify to the influence of dataset size and model capacity for the ELBO's OOD detection performance in the 2D case and practical image dataset pairs. We found that the prior is still improper and the overestimation issue still happens.
> We would also add some discussions about similar empirical findings in previous papers like distributional matching [1], which has been included in our paper, and paper [2] to make it more relevant to practical cases.
>
> [1] Mihaela R., et al.. Distribution matching in variational inference.
>
> [2] Dai, B., et al.. Diagnosing and Enhancing VAE Models.
>
> **For weakness 2:**
> The design of the calibration term is to mitigate the entropy gap between ID and OOD datasets, as shown in Eq.8.
> The reason why we use image complexity (evaluated by image compression methods like SVD) to approximate the dataset entropy is inspired by the previous work [13].
> The high-level idea is that an existing dataset, such as MNIST, always contains similar data points (similar content and texture complexity), which leads to similar image complexity, and datasets with more complex texture should have greater entropies, which is in line with the image complexity, e.g., FashionMNIST's entropy is reasonable to be greater than MNIST and the FashionMNIST's image complexity is shown to be higher as shown in Figure 5. However, since image complexity is not equal to entropy in scale, we propose the scaled version to scale the image complexity to approximate dataset entropy in Line 246 and Appendix D.
>
> SVD would fail when the datasets are similar in image complexity like CIFAR-10(ID)/CIFAR-100(OOD) compared with other dataset pairs as shown in Appendix G.
>
> We will add the explanation of "image complexity" and use it consistently.
>
> **For weakness 3**: Sorry for the misunderstanding.
> 1) we don't bold "WAIC" because we have claimed in Table 1 that "best results achieved
> by the methods of the category “Not ensembles” of “Unsupervised” have been bold".
>
> 2) "performance of a standard VAE" is the performance of the "ELBO [25]" in all tables as stated in line 271 "compare our method with a standard VAE [25]". We will make it more clear in the main paper.
>
> 3) The density plot of Fig. 6 (b) is the ablation study's visualization of PHP method (its auroc is 89.7%) instead of the AVOID (its auroc is 99.2%). AUROC is a commonly used threshold-free metric, named area under the receiver operating characteristic curve, which is not the threshold-needed "accuracy". PHP and AVOID's ROC curves are shown in **Figure 1 \(d\) of the one-page rebuttal pdf**.
>
> 4) Thanks for this helpful suggestion! The standard deviations have been included in Appendix H and we would add the standard derivation to the main paper's tables.
>
> 5) The CIFAR-10(ID)/CIFAR-100(OOD) dataset pair is a commonly used "hard task" pair in OOD detection tasks [1-5], since CIFAR-100 contains many unseen categories in CIFAR-10 and the image style and texture is pretty similar.
>
> [1] Morningstar, W., et al. Density of states estimation for out of distribution detection.
>
> [2] Nalisnick, E., et al. Detecting out-of-distribution inputs to deep generative models using typically.
>
> [3] Serrà, J., et al. Input Complexity and Out-of-distribution Detection with Likelihood-based Generative Models.
>
> [4] Cao, S., and Zhongfei Z. Deep hybrid models for out-of-distribution detection.
>
> [5] Fort, S., Jie R., and Balaji L. Exploring the limits of out-of-distribution detection.
>
> 6) Thanks for the helpful suggestion and we will improve the expression.
>
> **For question 1:** Sorry for the confusion. As shown in Figure 4, the sentence means that the deep-blue points (latent representation $z\sim q(z|x)$ of FashionMNIST) are much more distinguishable from the red points (MNIST) than the light-blue points (latent $z$ sampled from $\mathcal{N}(0,I)$) from the red points.
>
> **For question 2:** Actually, it would be used to alleviate the overestimation caused by factor 2, i.e., dataset entropy.
>
> Yes, we have the same thoughts as you that it could be better to balance the weights between the regularizer (calibration term) and the remainder of the objective, and we have already provided a scaled version of the calibration function in Section 4.3, whose scale score is adaptive on the entropy of ID dataset.
>
> **For question 3:** Sorry for the misunderstanding. The LSTM-based prior is learned after the VAE is trained instead of end-to-end training with the VAE together. VAE with a more complicated prior may help alleviate the overestimation problem but we think it is not efficient and not compatible with existing VAEs which are typically equipped with the efficient reparameterization trick based on a standard Gaussian prior.
>
> **For question 4:** Thanks for recommending this work. We will add the discussion of this paper to the paper since this work firstly provides an interesting and novel perspective from the gradients in doing OOD detection. But we find the performance of this method (with batch size B=1) is worse than our method and some SOTA VAE-based baselines, e.g., the AUROC in CIFAR-10(ID)/SVHN(OOD) is only 0.82. The superiority of this method is demonstrated when increasing the batch size to 5, but it is not suitable to our method and experimental setting.
>
> **For limitations:** See the last paragraph in the Author Rebuttal for all reviewers.

---

> > ### Comment · Reviewer_LEk9 · 2023-08-10
> > **Response to rebuttal**
> >
> > * On weakness 1: I do not see how the conclusion in 180-183 " can be flexibly extended for other non-linear VAEs", please explain this more, I would like to find an agreement on this point. (9-11) states a linear VAE. Where do you show its extension to non-linear VAEs?
> > * On weakness 2: Thank you for the explanation! To me, this term is yet not well motivated, and relies too much on it "being used in previous work". Also, it seems to be highly limited to image data, which limits the scope of the method.
> > * On weakness: The explanations are very helpful, and should be used to clarify the experimental results in the paper. With these changes, and with the new results on standard deviations, the experimental results are improved.
> > * The questions are well answered.

---

> > > ### Author Response · Authors · 2023-08-14
> > > **Clarification of the "extension" and the motivation of the DEC method**
> > >
> > > We deeply appreciate your valuable feedback, which could significantly improve the quality of our paper. We hope our response below could adequately address your concerns and if there are any remaining concerns, we are more than willing to engage in further discussion with you.
> > >
> > > ---
> > > **Extension to other non-linear VAEs:**
> > >
> > > Sorry for the confusion about the statement "the conclusion can be flexibly extended for other non-linear VAEs", we feel like the confusion may be due to the unclear organization of section 3.2 and the unclear previous response.
> > > Without access to modify it now, please allow us to state the pipeline of section 3.2:
> > > * Part 1: "When the design of the prior is proper?": For a single-modal Gaussian data distribution, a linear VAE (stated in Eq. (9)) with its optimal parameters can satisfy $q(z)=p(z)$, where the overestimation issue will not occur.
> > >
> > >
> > > * Part 2: "When the design of prior is NOT proper?": We found that, for a multi-modal Gaussian data distribution, a linear VAE (same as stated in Eq. (9)) with its optimal parameters, which can still be obtained analytically, can **not** satisfy the condition $q(z)=p(z)$, which indicates the design of the prior is not proper and will lead to overestimation.
> > >
> > >
> > > * Part 3: "More empirical studies on the improper design of prior": Considering the fact that more VAEs in practice are non-linear, we try to investigate the overestimation of non-linear VAEs. However, it is hard and meaningless to get the analytical solution of the parameters for a single specific non-linear VAE, which inspires us to conduct a series of empirical studies on the cases of non-linear VAEs. Note that, we have included the results of non-linear VAEs with various network structures in Fig.3 ((a-b) is an MLP-based VAE for a synthesized 2D multi-modal dataset, and (c-d) is a CNN-based VAE for realistic image datasets), and can find that the condition $q(z)=p(z)$ can hardly be achieved because **it is difficult or even impossible to make $q_\phi(z|x)$ to be exactly the same as $p(z|x)$** through optimization in practice [1,2,3].
> > >
> > >     [1] Mihaela R., Balaji L., and Shakir M. Distribution matching in variational inference.
> > >
> > >     [2] Dai, B., and David W. "Diagnosing and Enhancing VAE Models."
> > >
> > >     [3] Dai, Bin, Li Kevin Wenliang, and David Wipf. "On the Value of Infinite Gradients in Variational Autoencoder Models."
> > >
> > > Based on part 3's findings and analysis, a **conclusion** is that these non-linear VAEs used in our paper, which are trained by optimizing ELBO, cannot satisfy the condition $q(z)=p(z)$ and lead to overestimation in practice. Please note that the related findings in paper [1,2,3] are also not specific to the neural network setting for modeling $q_\phi(z|x)$. Thus, **this conclusion** can be **flexibly "extended" for other non-linear VAEs** with different network structures (e.g., number of layers and network types), which helps us to achieve the conclusion in lines 180-183.
> > >
> > >
> > >
> > >
> > > ---
> > > **Motivation of DEC:**
> > >
> > > For the motivation of DEC, we also discussed it with reviewer GLsZ (please see https://openreview.net/forum?id=31zVEkOGYU&noteId=LXigKkXvo8), where we found there is a big misunderstanding of the relationship between image complexity and dataset entropy. Briefly, no matter what the relationship between image complexity and dataset entropy is, it won't affect the effectiveness of our DEC method. Because **we don't use image complexity to estimate dataset entropy** in the original paper (Sorry for our last non-rigorous response).
> > >
> > >
> > > The main contribution of our paper is to identify the root cause of the overestimation issue of a trained VAE and **SVD-based method is only one of the choices** of implementing the dataset entropy calibration (DEC) method. We choose the SVD-based DEC method because the **benchmarks** of the OOD detection mainly focus on the image datasets and the SVD-based DEC method could make it easier to be understood and testify.
> > > We believe that there will be other ideal implementation methods to provide more accurate discrimination between ID and OOD, and further improve the performance of OOD detection.
> > > But please note that factor 2 (higher ID dataset entropy)'s contribution to overestimation is **not restricted to the data type**.
> > >
> > > Besides, SVD could be applied to other "matrix" data types **beyond just image data**. High-dimension non-matrix data could also be transformed to be a matrix and then apply the SVD method. For low-dimension data, there could be some traditional density estimation methods like the Gaussian mixture model that can be helpful. Note that VAE and other Deep Generative Models are usually focused on high-dimension data.
> > >
> > > We deeply regret not having provided a clear clarification regarding the "extension" and "motivation of DEC" in our initial response. We hope that this follow-up response will adequately address your concerns.
> > >
> > > [A further discussion could be found here: https://openreview.net/forum?id=31zVEkOGYU&noteId=QSTNNYppKw ]

---

> > > ### Author Response · Authors · 2023-08-18
> > >
> > > Dear reviewer LEk9,
> > >
> > > We deeply appreciate the valuable time and effort you've taken to share your valuable insights and comments.
> > > Since your thoughts are of great importance to us, could we kindly request a small portion of your time to review our secondary response? We hope our response has clarified the meaning of "extension to other non-linear VAEs", the motivation of the developed method, and also its contribution to the field of unsupervised OOD detection. We are sincerely waiting for your feedback.
> > >
> > > Best regards,
> > >
> > > Authors

---

> > > ### Author Response · Authors · 2023-08-22
> > > **Thanks for your efforts and contributions during the discussion of the rebuttal!**
> > >
> > > Dear reviewer LEk9,
> > >
> > > We would like to express our sincere gratitude once again for your efforts and contributions during the discussion of the rebuttal, which has significantly improved the quality of our paper.
> > >
> > > Upon realizing that the discussion seems to be able to continue beyond the rebuttal deadline, we kindly request a brief extension of your time to assess our responses to your remaining concerns. We found both concerns are due to some misunderstandings. Should any concerns persist, we are more than eager to engage in further discussion with you. Thanks again!
> > >
> > > Best regards,
> > >
> > > Authors

---

### Official Review · Reviewer_Bzqi · 2023-07-06

**Soundness:** 3 good
**Presentation:** 3 good
**Contribution:** 3 good
**Rating:** 6
**Confidence:** 4

**Summary:**

The paper studies unsupervised OOD detection (i.e., training data contains no labels) using deep generative models. DGMs model the probability distribution of the inputs, and can be an ideal candidate for unsupervised OOD detection. The authors study one specific class of DGMs, namely VAEs. They show that VAEs suffer from overestimation problem ($P(x_{ood}) > P(x_{id})$) due to two main reasons — dataset’s inherent entropy and improper design of prior distribution. The paper then proceeds to theoretically suggest ways to mitigate this issue, and shows experimental results that do so.

**Strengths:**

1. The theory of the paper is simple but inspiring, and matches neatly with the designed algorithm.
2. The experiments are well-designed and executed.
3. The ablation studies are well-done.

**Weaknesses:**

1. Prior work such as [1] that discusses causes of deep generating models’ (specifically, normalizing flows) reason for failure to perform OOD detection was not cited/discussed in the paper. Similarly, [2] is also an important paper for using DGMs for OOD detection that wasn’t cited.
2. The paper is not self-contained and the organization could be improved — for example, one could put the limitations in the main paper instead of in the appendix.
3. Notation of the paper. For example, $p(x) = N(x | 0, \Sigma_x)$ can be more readable as $x \sim N(0, \Sigma_x)$, following more commonly used convention.

[1] Polina Kirichenko, Pavel Izmailov, Andrew Gordon Wilson. Why Normalizing Flows Fail to Detect Out-of-Distribution Data, https://arxiv.org/abs/2006.08545, 2020

[2] Eric Nalisnick, Akihiro Matsukawa, Yee Whye Teh, and Balaji Lakshminarayanan. Detecting out-of-distribution inputs to deep generative models using a test for typicality. arXiv preprint arXiv:1906.02994, 2019.

**Questions:**

1. What is the architecture used for the supervised OOD detection methods in Table 1? Is it a 3 layer MLP?
2. Intuitively, supervised OOD detection methods should do better than unsupervised ones, since having access to more information should never hurt the performance. However, that is not the case in Table 1. Do the authors have an explanation for this?
3. Line 278, why is the max epochs set to such a high value (1000)? Is there some sort of early stopping that is used here?
4. I am curious on how the reasons noted by [1], namely normalizing flows learning latent representations based on local pixel correlations and not semantic content, are relevant for VAEs. Does VAEs focus on semantic content and not local pixel correlations?


[1] Polina Kirichenko, Pavel Izmailov, Andrew Gordon Wilson. Why Normalizing Flows Fail to Detect Out-of-Distribution Data, https://arxiv.org/abs/2006.08545, 2020

---

> ### Author Rebuttal · Authors · 2023-08-10
>
> **For weakness 1:** Thanks for your suggestion and we will add these citations.
>
> **For weakness 2:** We're sorry for the non-self-contained organization and we will add the limitation to the main paper.
>
> **For weakness 3:** Thanks for your comment on the notation and we will improve it as you suggested.
>
> **For question 1:** For the supervised OOD detection methods, the performance is based on the network structures described in their original papers instead of a 3-layer MLP. Most of the supervised methods are based on a classifier, e.g., LN [9] is based on WRN-40-2 [1].
>
> [9] Wei, Hongxin, et al. "Mitigating neural network overconfidence with logit normalization." International Conference on Machine Learning. PMLR, 2022.
> [1] Zagoruyko, Sergey, and Nikos Komodakis. "Wide residual networks." arXiv preprint arXiv:1605.07146 (2016).
>
> **For question 2:** This is not the truth. In the research area of supervised OOD detection, there is a common issue: neural networks are known to suffer from the overconfidence issue, where they produce abnormally high confidence for both in- and out-of-distribution inputs [9]. In other words, the classifier would abnormally assign wrong categories to OOD data. This is still a hard issue to be addressed by the supervised methods. We will add this explanation to the main paper.
>
> **For question 3:** We follow the same experimental setting with these SOTA VAE-based baselines (HVK [17] and $\mathcal{LLR}^{ada}$ [18]), where the max epoch is set as 1000 and the best neural network parameters are saved according to the best ELBO in the training set. For the early stopping, we found actually the ELBO is stable after 100~200 epochs with our model trained in FashionMNIST and CIFAR-10 (see the training curve of ELBO in **Figure 1 \(c\) of one-page rebuttal pdf**), which could be considered set as a proper early stopping epoch.
>
> **For question 4:**
> As the cause of the failure analyzed in paper [1], the flow model's learned local pixel correlations and generic image-to-latentspace transformations are not specific to the training image dataset. Since VAEs also focus on both semantic content (mainly in the image-to-latentspace process by the encoder $q_\phi(z|x)$) and local pixel correlations (mainly in the reconstruction process by decoder $p_\theta(x|z)$), similar phenomenon has been observed in VAEs: HVK [17] and $\mathcal{LLR}^{ada}$ [18] found that the reconstruction for ID and OOD data could be both high-quality, which means VAEs also learned some information that is not specific to the ID training dataset, i.e., low-level information instead of semantic information. Part of the success of the paper [1], HVK[17], and $\mathcal{LLR}^{ada}$ [18] are the same: forcing the model to learn ID training dataset's specific semantic information. In our paper, that is the PHP method, which learns the $q(z)=\int q_\phi(z|x)p(x)$ and then uses $D_{KL}[q(z|x)||q(z)]$ instead of $D_{KL}[q(z|x)||p(z)]$, since $p(z)$ is uninformative and could contain non-training-dataset information.

---

> > ### Comment · Reviewer_Bzqi · 2023-08-14
> >
> > This is an interesting paper overall, and I thank the authors for writing a comprehensive rebuttal to my concerns. The idea of decomposing the ELBO and addressing concerns of over-estimation is neat and ties the method of the paper nicely.
> >
> > Some of the citations [1-9] are missing. A few more comments/questions:
> >
> > 1. **(class conditioning)** The method is unsupervised in the sense that it does not use the in-distribution labels. To add to the comment by the authors on unsupervised vs supervised OOD detection, could learning class conditional distributions on the data help even more with OOD detection? i.e., could this paper's method be generalized where we can use the class labels?
> >
> > 2. **(weak baselines)** The paper compares with outlier exposure [1] as a supervised method using auxiliary data. However, outlier exposure, despite being a foundational method in this field, was published in 2019 and several direct improvements over it has been found. For example, energy based out-of-distribution detection [2] can be a stronger baseline. Same thing goes for supervised methods not using auxiliary data, for example, the authors compare against CP, which is the MSP method from [3]. However, energy score [2] and MaxLogit [4] has been known to perform better.
> >
> > 3. **(Energy score)** Is EN in table 1 the same as energy score method? It cites the same paper.
> >
> > 4. **(Error bars in table 1 or 2)** The paper does not report error bars in the table. While Appendix H does mention average error bars, it is unconventional **to the best of my knowledge**, and individual comparisons without associated error bars is meaningless.
> >
> > 5. **(Comparison to transductive setting)** Since the method uses $n_{id}$ and $n_{x}$ for test example $x$ to calculate the calibration term $C(x)$, it has similarities to transductive/semi-supervised methods that use auxiliary datasets containing OOD examples. Comparisons to the settings, ideas and results for ERD and binary classifier [5], WOODS [6], DCM (Transductive setting) [7], where the test dataset is used to update the models/scores and get improved OOD detection performance would be important for the paper.
> >
> > 6. **(Over-estimation in supervised OOD detection)** The authors mention the overconfidence issue on supervised learning setup as well, and cite [8]. I think mentioning this in the paper itself, drawing parallels between the unsupervised and supervised case, and methods that people use for mitigating the overconfidence issue in the supervised case [7][8], would be important.
> >
> > 7. **(ID/OOD data pair construction)** The authors use CIFAR-10 and STL-10 as an ID/OOD data pair. This blurs the line of ID vs OOD, i.e., what is the definition of OOD data? I think it is common to take datasets that do not contain any common classes between them as (ID, OOD) pair, see the dataset construction in [9]. Reviewer LEk9 also complains about this issue, and while CIFAR-10 and CIFAR-100 have non-overlapping classes and is often considered a hard-OOD task, I would argue CIFAR-10 and STL-10, sharing 9 out of 10 classes between them, should not suffice as an (ID, OOD) paper. This also brings the question of **what is OOD detection** as asked by [9]. The practical use of OOD detection is to have a sort of conservative deferring mechanism [1, 7], where given a test example x, one either makes a classification on it, or in case it does not belong to any of the K classes the model knows, it is deferred to an expert. While this no longer works in this paper's unsupervised setting, consider the image of a cat from CIFAR-10, $x_1$, and the image of a cat from STL-10, $x_2$. Other than resolution characteristic (32 x 32 for $x_1$ vs 96 x 96 for $x_2$) why should $x_1$ be classified as ID and $x_2$ as OOD?
> >
> > **Based on all these questions/comments, I am keeping my original score.**
> >
> >
> > [1] Deep Anomaly Detection with Outlier Exposure, https://arxiv.org/abs/1812.04606
> >
> > [2] Energy-based Out-of-distribution Detection, https://arxiv.org/abs/2010.03759
> >
> > [3] A Baseline for Detecting Misclassified and Out-of-Distribution Examples in Neural Networks, https://arxiv.org/abs/1610.02136
> >
> > [4] Scaling Out-of-Distribution Detection for Real-World Settings, https://arxiv.org/abs/1911.11132
> >
> > [5] Semi-supervised novelty detection using ensembles with regularized disagreement, https://arxiv.org/abs/2012.05825, Official implementation: https://github.com/ericpts/ERD
> >
> > [6] Training OOD Detectors in their Natural Habitats, https://proceedings.mlr.press/v162/katz-samuels22a/katz-samuels22a.pdf, Official implementation: https://github.com/jkatzsam/woods_ood
> >
> > [7] Conservative Prediction via Data-Driven Confidence Minimization, https://arxiv.org/abs/2306.04974, Official implementation: https://github.com/tajwarfahim/dcm
> >
> > [8] Mitigating Neural Network Overconfidence with Logit Normalization, https://arxiv.org/abs/2205.09310
> >
> > [9] No True State-of-the-Art? OOD Detection Methods are Inconsistent across Datasets, https://arxiv.org/abs/2109.05554

---

> > > ### Author Response · Authors · 2023-08-18
> > > **Further response (part 1/2)**
> > >
> > > Thanks for your valuable feedback! We hope our response below could adequately address your concerns and if there are any remaining concerns, we are more than willing to engage in further discussion with you.
> > >
> > > ---
> > >
> > > **ID/OOD data pair construction:** We want to answer this question regarding "what is OOD detection" in unsupervised OOD detection at first. The OOD data in the unsupervised OOD detection task can arise not only due to semantic differences (category information) but also from statistical differences. For example, a vertically flipped (VFlip) in-distribution testing sample is also considered as an OOD sample for the in-distribution testing sample itself [1, 2], though they share the same category information. And we also add experiments on the "VFlip" case in the discussion with the reviewer ogin (https://openreview.net/forum?id=31zVEkOGYU&noteId=unI5jvIInm), where the experiments could support our analysis of the latent factor and effectiveness of the PHP method.
> > >
> > > [1] Choi, Hyunsun, Eric Jang, and Alexander A. Alemi. "WAIC, but Why? Generative Ensembles for Robust Anomaly Detection."
> > >
> > > [2] Morningstar, Warren, et al. "Density of states estimation for out of distribution detection." International Conference on Artificial Intelligence and Statistics. PMLR, 2021.
> > >
> > > **class condition:** Yes, the paper's method could be flexibly generalized when the class labels are available. Since the class label contains rich semantic information, it could be used to provide more expressive latent data representations and could further improve the OOD detection performance of our PHP method.
> > >
> > >
> > > **weak baselines:** Thanks for your helpful suggestion and we would add the results of these methods to Table 1. Since our method focuses on unsupervised OOD detection, we cite most of the results directly from the reported results in two SOTA unsupervised OOD detection baselines [3, 4]. We would carefully go through the results in Table 1 and update it with the latest performance for methods in the category of the "supervised" and "auxiliary". We note that the results in the category "unsupervised" is already the SOTA performance and share the same experimental setting as our method.
> > >
> > >
> > >
> > > **Energy score:** Yes, the performance of this energy-based method could achieve very good performance in the CIFAR(ID)/SVHN(OOD) pair (could even achieve 99.41% at the setting of applying fine-tuning with WideResNet as shown in their original paper), where we also directly cite the result reported in [4].
> > >
> > > [3] Havtorn, Jakob D., et al. "Hierarchical vaes know what they don’t know." International Conference on Machine Learning. PMLR, 2021.
> > >
> > > [4] Li, Yewen, et al. "Out-of-distribution detection with an adaptive likelihood ratio on informative hierarchical vae." Advances in Neural Information Processing Systems 35 (2022): 7383-7396.
> > >
> > > **Error bars in Table 1 or 2**: Thanks for your recommendation of adding individual comparisons with associated error bars for Table 1 or 2.
> > > We are very sorry that we did not have enough time to provide a detailed Table 2 in the first rebuttal stage. We choose Table 2 since the unsupervised OOD detection baselines including ELBO, HVK, and $\mathcal{LLR}^{ada}$ are the most related baselines to our methods. To obtain these error bars, all the experiments have been conducted 5 times with random seeds from 1 to 5. Note that, the DEC method is an SVD-based method containing no stochasticity.
> > > From the experimental results shown in the tables, all these unsupervised OOD detection methods are relatively stable.
> > >
> > >
> > >
> > > | FashionMNIST(ID) / MNIST(OOD) | | ||
> > > | --- | ------ | ------- | ----- |
> > > | Method | AUROC $\uparrow$ | AUPRC $\uparrow$ | FPR80 $\downarrow$|
> > > | ELBO [25] | 23.5 $\pm$ 0.820 | 35.6 $\pm$ 0.859 | 98.5 $\pm$ 0.389 |
> > > | HVK [17] | 98.4 $\pm$ 0.798  | 98.4 $\pm$ 0.734  | 1.3 $\pm$ 0.042 |
> > > | $\mathcal{LLR}^{ada}$ [18] | 97.0 $\pm$ 0.583 | 97.6 $\pm$ 0.723 |0.9 $\pm$ 0.039 |
> > > | -ours | | | |
> > > | PHP  | 89.7 $\pm$ 0.548 | 90.3 $\pm$ 0.507 | 13.3 $\pm$ 0.249  |
> > > | DEC | 34.1 $\pm$ 0.000 | 40.7 $\pm$ 0.000 | 92.5 $\pm$ 0.000 |
> > > | AVOID | 99.2 $\pm$ 0.516  | 99.4 $\pm$ 0.605  | 0.0 $\pm$ 0.009 |
> > >
> > >
> > > |     CIFAR-10(ID) / SVHN(OOD) | | ||
> > > | --- | ------ | ------- | ----- |
> > > | Method | AUROC $\uparrow$ | AUPRC $\uparrow$ | FPR80 $\downarrow$|
> > > | ELBO [25] | 24.9 $\pm$ 1.418    | 36.7 $\pm$ 1.522   | 94.6 $\pm$ 0.965  |
> > > | HVK [17] | 89.1 $\pm$ 2.323  | 87.5 $\pm$ 2.967  | 17.2 $\pm$ 2.005 |
> > > | $\mathcal{LLR}^{ada}$ [18] | 92.6 $\pm$ 0.411 | 91.8 $\pm$ 0.542 | 11.1 $\pm$ 0.277 |
> > > | -ours | | | |
> > > | PHP  | 39.6 $\pm$ 1.379  | 42.6 $\pm$ 1.533  | 85.7 $\pm$ 0.691   |
> > > | DEC | 87.8 $\pm$ 0.000  | 89.9 $\pm$ 0.000  | 17.8 $\pm$ 0.000  |
> > > | AVOID | 94.5 $\pm$ 1.440   | 95.3 $\pm$ 1.487   | 4.24 $\pm$0.365 |

---

> > > > ### Author Response · Authors · 2023-08-18
> > > > **Further response (part 2/2)**
> > > >
> > > > **Comparison to transductive setting:**
> > > > Thanks for this suggestion! After reading the recommended papers and their corresponding codes, we found these methods [5,6,7] are focused on exploiting an (unlabelled) auxiliary dataset to improve the OOD detection performance based on a classifier, which needs to incorporate the in-distribution training set's labels during the training of classifier that is different from the settings of unsupervised cases. More specifically, ERD [5] is based on ResNet and WideResNet, WOODS [6] is based on a WideResNet, and DCM [7] is based on ResNet and WideResNet. However, our work is focused on analyzing and alleviating the VAE's overestimation issue in the unsupervised OOD detection task, which is more related to the DGM-based methods. Besides, our DEC method, using $n_{id}$ from the in-distribution training set and computing $n_x$ for each unseen testing sample, does not need the incorporation of an auxiliary dataset.
> > > >
> > > > Though the setting in [5, 6, 7] is not the same as ours, their idea of introducing an auxiliary dataset is still an interesting and potential direction to improve unsupervised OOD detection. As analyzed in DCM [7], if the auxiliary dataset includes samples from the OOD region of interest, exploiting the dataset could be provably helpful to separate ID and OOD inputs. Thus, constructing an auxiliary dataset that covers most of the OOD data region of interest is promising to enhance the DGM's OOD detection performance on certain OOD data types. Although this is out of the scope of our paper, we still feel very appreciative of your suggestion and would keep focusing on this direction.
> > > >
> > > >
> > > >
> > > > **Over-estimation in supervised OOD detection:**
> > > > Thanks for your suggestion!
> > > > Without access to modify the main paper now, we present a discussion here about  the methods that people use for mitigating the overconfidence issue in the supervised case (especially the classifier-based methods), which could be divided into two categories, and draw parallels between the unsupervised and supervised cases:
> > > > 1) Designing a score function based on the properties of a **trained** classifier, such as maximum softmax probability [5], Mahalanobis distance-based score [6], energy-based score [7], and GradNorm score [8]. These methods operate on the premise that the statistical characteristics exhibited by the classifier when presented with an in-distribution (ID) data example are distinct from those observed when handling an out-of-distribution (OOD) data example. For unsupervised case, without the label to train a classifier, it is similar to designing a score function based on a trained deep generative model (DGM), such as HVK [3] that exploit the relationship between the posterior and prior latent distribution existed in a trained VAE to do OOD detection. More methods could be seen in lines 529-538 of Appendix B.2. Our method also focuses on a trained VAE, which has the advantage of "plug-and-play" and could be compatible with the existing VAEs.
> > > > 3) Introducing regularization techniques during the classifier's **training phase**, such as adding a fix to the cross-entropy loss [9], encouraging the classifier to give predictions with uniform distribution for OOD data [10], and shaping the log-likelihood by energy-based regularizer [11]. After training with the regularizer, the classifier exhibits differing statistics between ID and OOD data. There are also some similar works that modify the training objective of the DGM in unsupervised cases, e.g., the $\mathcal{LLR}^{ada}$ [4] adds a partial generation term into the ELBO during training a VAE that could explicitly enhance the semantic information's quality in the latent variables that could be helpful for OOD detection. More related works could be seen in lines 517-529 of Appendix B.2. However, these methods avoid analyzing the root cause of the original DGMs' overestimation issue.
> > > >
> > > > [3] Havtorn, Jakob D., et al. "Hierarchical vaes know what they don’t know." ICML, 2021.
> > > >
> > > > [4] Li, Yewen, et al. "Out-of-distribution detection with an adaptive likelihood ratio on informative hierarchical vae." NeurIPS, 2022.
> > > >
> > > > [5] Hendrycks, D., et al. "A baseline for detecting misclassified and out-of-distribution examples in neural networks."
> > > >
> > > > [6] Lee, K., et al. "A simple unified framework for detecting out-of-distribution samples and adversarial attacks." NeurIPS, 2018.
> > > >
> > > > [7] Morteza, P., et al. "Provable guarantees for understanding out-of-distribution detection." AAAI, 2022.
> > > >
> > > > [8] Huang, R., et al. "On the importance of gradients for detecting distributional shifts in the wild." NeurIPS, 2021.
> > > >
> > > > [9] Wei, Hongxin, et al. "Mitigating neural network overconfidence with logit normalization." ICLR, 2022.
> > > >
> > > > [10] Hendrycks, D., et al. "Deep anomaly detection with outlier exposure." ICLR, 2019.
> > > >
> > > > [11] Katz-Samuels, J., et al. "Training ood detectors in their natural habitats." ICML, 2022.

---

> > > > > ### Comment · Reviewer_Bzqi · 2023-08-18
> > > > >
> > > > > I thank the authors for providing a thorough and thoughtful reply to my feedback.
> > > > >
> > > > > **(Error bars)** Thanks for providing the error bars here. I would request the authors to include detailed error bars for as many methods as possible (or discuss why this is not possible) in the next iteration of the paper.
> > > > >
> > > > > **(ID/OOD data pair construction)** The authors raise a good point here: since class information is not available in unsupervised OOD detection, the distinction between ID and OOD data is characterized by statistical difference and not only semantic difference. This is a good point, and I would request adding this explanation to the main paper. I had a misunderstanding here, and I thank the authors for clarifying this.
> > > > >
> > > > > **(class condition)** Could this be mentioned as a future work/limitation in the paper?
> > > > >
> > > > > **(Comparison to transductive setting)** I agree that a comparison to the transductive setting same as ERD, binary classifier and DCM is not directly relevant for the paper. However, the authors do mention baselines that use auxiliary dataset such as Outlier exposure, for the sake of completeness, I would ask to add these other baselines in the paper (table 1) as well.
> > > > >
> > > > > **Overall the authors have provided a concise and sufficient address to my feedback/concerns which have clarified the paper, and I raise my score from 5 to 6. I would request the authors to add these changes suggested to the final version of the paper.**

---

> > > > > > ### Author Response · Authors · 2023-08-19
> > > > > > **We are very delighted that we have addressed your concerns!**
> > > > > >
> > > > > > We are very delighted that we have addressed your concerns! Thanks for your further suggestions and we will add the suggested changes to the final version of the paper!
> > > > > >
> > > > > > **error bars:** Thanks for your suggestion, we would include detailed error bars for as many methods as possible in the next iteration of the paper.
> > > > > >
> > > > > > **ID/OOD data pair construction:** We are happy that we could address this concern. We would add this explanation to the main paper, too.
> > > > > >
> > > > > > **class condition:** Yes, we would add this class condition as a part of the future work/limitation in the paper.
> > > > > >
> > > > > > **comparison to transductive setting:** Thanks for your suggestion, we would add these other baselines to Table 1 in the paper.
> > > > > >
> > > > > > Thanks again for your continued engagement in the discussion, which could significantly contribute to the improvement of the quality of our work!

---

### Official Review · Reviewer_ogin · 2023-07-07

**Soundness:** 3 good
**Presentation:** 3 good
**Contribution:** 2 fair
**Rating:** 5
**Confidence:** 4

**Summary:**

In the context of VAE, the authors identified two factors that potentially cause VAE to assign higher likelihood to OOD data than ID data. They propose a new scoring mechanism that improves upon VAE's overestimation of the likelihood on OOD samples.

**Strengths:**

- Decomposing the ELBO carefully is interesting. In particular, they give a new prior design targeting the overestimation issue.
- They have a scoring method that improves upon the standard ELBO, which partially validifies their analysis.

**Weaknesses:**

- the derivation assumes the model distribution can converge exactly to the true one, but this is impractical.

If it does, there should be no overestimation issue to begin with (for practical datasets that are arguably separable, e.g. SVHN vs CIFAR). Moreover, even if it is possible in theory, the empirical and theoretical observations in [1, 2] will prevent this from happening in practice.

If it doesn't, the derivation will leave an error gap that is not analyzed. In short, the key reasoning in the above is that real distribution is often supported on low dimensional sets, while model distribution is fully supported.

- the evaluation is a bit outdated on easier benchmarks. To solidifies AVOID's practical impact, evaluation on the harder tasks as in DoSE [3] is necessary.


[1] Dai, Bin, and David Wipf. "Diagnosing and Enhancing VAE Models." International Conference on Learning Representations. 2018.

[2] Dai, Bin, Li Kevin Wenliang, and David Wipf. "On the Value of Infinite Gradients in Variational Autoencoder Models." Advances in Neural Information Processing Systems. 2021.

[3] Morningstar, Warren, et al. "Density of states estimation for out of distribution detection." International Conference on Artificial Intelligence and Statistics. PMLR, 2021.

**Questions:**

I'm happy to miss something in the paper, and be corrected. See the weakness above.

---

> ### Author Rebuttal · Authors · 2023-08-10
>
> Thanks for your insightful comments, the following is our response.
>
> **For weakness 1:** We absolutely agree with your point that the model distribution can hardly converge to the data distribution in practical VAEs, and there does exist a third term in ELBO to affect the performance of ELBO-based OOD detection. However, we need to claim that this error gap can only be alleviated **during the training** of VAEs but our paper focuses on improving the OOD detection performance of VAEs **after training** via simple and universal methods, which are agnostic to the training scheme or model architecture of different VAEs.
>
> Moreover, the derivation in our paper is for readers to easily understand our method (note that the analytical posteriors of $p_\theta(z|x)$ can be obtained in some Gaussian cases to make the model distribution equal to the data distribution) and we have provided a more rigorous derivation in the following to clarify your concerns (the original derivation has been included in Appendix C.1).
>
> The relationship between $p_{\theta}(x)$ and $\text{ELBO}(x)$ is:
> $$\log p_\theta(x)=\mathbb{E}\_{z\sim q_\phi(z|x)}[\log p_\theta(x|z)]-D_{KL}[q_\phi(z|x)||p(z)]+D_{KL}[q_\phi(z|x)||p(z|x)]=\text{ELBO}(x)+D_{KL}[q_\phi(z|x)||p(z|x)].$$
>
> Assuming that the ground truth of the ID data distribution is $p(x)$, and if we expect $\text{ELBO}(x)$ to converge exactly to $p(x)$ (where we'd acknowledge no overestimation issue), then two assumptions must be satisfied:
> 1) the encoder $q_\phi(z|x)$ should make $D_{KL}[q_\phi(z|x)||p(z|x)]=0$;
> 2) the decoder $p_\theta(x|z)$ should make $p_\theta(x)=\int p_\theta(x|z)p(z)dz=p(x)$.
>
> We strongly agree that these assumptions are hard to achieve but our methods are **NOT** based on these two assumptions, i.e., our methods are focusing on a trained VAE, and the analyzed two factors' contribution to overestimation is **NOT** affected by the two assumptions, though it does have an error gap to be analyzed.
>
> To be more rigorous, a **trained** VAE has the following property (we will correct the corresponding equations in the main paper):
> $$\mathbb{E}_{x{\sim}{p(x)}}[\text{ELBO}(x)] = -\mathcal{H}(x) - D\_{KL}[q(z)||p(z)] + \text{const}_p,$$
> where $\text{const}_p$ is a const that is only related to the $p(x)$ after finishing training the VAE and would not be changed by our PHP and DEC methods.  Now, we give a detailed derivation for $\text{const}_p$:
>
> For each term in $\mathbb{E}\_{x{\sim}{p(x)}}[\text{ELBO}(x)]$ we could have
> $$\mathbb{E}\_{x{\sim}{p(x)}}[\mathbb{E}\_{z{\sim}{q_\phi(z|x)}}\log{p_\theta(x|z)}]=\mathbb{E}\_{p(x)q_\phi(z|x)}[\log\frac{p_\theta(z|x)}{p(z)}p(x)]=\mathcal{I}\_{q,p}(x,z)-\mathcal{H}(x);$$
> $$\mathbb{E}\_{x{\sim}{p(x)}}[D_{KL}(q_{\phi}(z|x)||p(z))] = \mathbb{E}\_{p(x)q_{\phi}(z|x)}[\log\frac{q_\phi(z|x)}{q(z)}\frac{q(z)}{p(z)}]=\mathcal{I}\_q{(x,z)} + D_{KL}(q(z)||p(z)),$$
> where
> $$\mathcal{I}\_{q,p}(x,z)=-\mathcal{H}\_{q,p}(z|x)+\mathcal{H}\_{q,p}(z)=\mathbb{E}\_{p(x)q_\phi(z|x)}[\log p_\theta(z|x)]-\mathbb{E}\_{q(z)}[\log p(z)];$$
>
> $$\mathcal{I}\_{q}(x,z)=-\mathcal{H}\_{q}(z|x)+\mathcal{H}\_{q}(z)=\mathbb {E}\_{p(x)q_\phi(z|x)}[\log q_\phi(z|x)]-\mathbb{E}\_{q(z)}[\log q(z)].$$
>
> Further, we have
> $$\mathbb{E}\_{x{\sim}{p(x)}}[\text{ELBO}(x)] = [\mathcal{I}\_{q,p}(x, z) - \mathcal{I}\_q(x,z)] - ({\mathcal{H}(x) + D_{KL}(q(z)||p(z))}),$$ where $\mathcal{I}\_{q,p}(x,z)$ will gradually approach $\mathcal{I}\_q(x,z)$ in the process of optimizing ELBO. Importantly, when $\theta$ and $\phi$ are fixed after training, thus $[\mathcal{I}\_{q,p}(x, z) - \mathcal{I}\_q(x,z)]$ is a const, i.e.,
> $$\text{const}\_p = \mathcal{I}\_{q,p}(x, z) - \mathcal{I}\_q(x,z),$$ and would not be changed during applying PHP (replacing $D_{KL}[q_\phi(z|x)||p(z)]$ to $D_{KL}[q_\phi(z|x)||\hat{q}_{id}(z)]$, note that $I_q(x,z)$ is not related to $p(z)$) and DEC (add a calibration term $\mathcal{C}$) method.
>
> We admit that there is an error gap brought by $\text{const}\_p$ in ELBO to influence the performance of ELBO-based OOD detection, and it can hardly be optimized to zero because it is difficult to achieve the above two assumptions for a usual non-linear VAE.
> However, the influence of this term will become more and more slight with the introduction of more powerful neural networks and optimization algorithms, and it is also beyond the investigation scope of this paper.
> We emphasize that our method only focuses on how to alleviate VAE's overestimation issue and improve the OOD detection performance **after training**, and believe that the performance of our method can be further improved as long as ELBO can be better optimized during training.
>
> Performance of the additional experiments on "harder tasks" (weakness 2) could also empirically support the above analysis and demonstrate the effectiveness in alleviating overestimation of addressing factors 1 and 2 by our method.
>
> ---
> **For weakness 2:**
> Thanks for your bringing these hard tasks into our sight. The "hard tasks" in DoSE [3] refer to the ID/OOD dataset pairs:  1) FashionMNIST(ID) / MNIST(OOD); 2) CIFAR-10(ID) / SVHN (OOD); 3) CelebA(ID) / CIFAR-10/100(OOD); 4) CIFAR-10(ID) / CIFAR-100(OOD) as described in "dataset" part of Appendix C in the full DoSE paper ( https://www.alexalemi.com/publications/dose.pdf). As shown in Tables 1, 2, and 3 of the main paper, our experiments have already included the "harder" dataset pairs 1, 2, and 4. Thus, we add the experiments for the dataset pair "CelebA(ID) / CIFAR-10/100(OOD)" with the comparison between ELBO (standard VAE), two SOTA VAE-based methods (HVK, $\mathcal{LLR}^{ada}$), and our method (AVOID) in the **Table 1 of "one-page rebuttal pdf"**. Additionally, we testify our method in more datasets with the VAEs trained in CelebA in the **Table 2 of "one-page rebuttal pdf"**. The experimental results demonstrate the effectiveness of our method, which mitigates the phenomenon of overestimation with two potential factors in ELBO.

---

> > ### Comment · Reviewer_ogin · 2023-08-14
> > **The hard cases in DoSE refer to the VFlip or HFlip versions, not just the pairs**
> >
> > As the title suggests, please refer to Table 1 in DoSE carefully. VFlip and HFlip are vertical and horizontal flips of the original test data. So OOD differs from IID by only one latent factor. Not going through them carefully weakens your statements.

---

> > > ### Author Response · Authors · 2023-08-16
> > > **Misunderstanding is caused by that "simple" and "hard" tasks have been previously defined in the DoSE**
> > >
> > > **We sincerely apologize for our misunderstanding of the concept of the  "harder task."** Actually, it is exactly due to that **we are too "carefully"** focused on going through the DoSE paper, where the "simple" and "hard" have been clearly defined in DoSE paper. We directly cite the original description below in their AISTATS'21 paper's page 12 (https://www.alexalemi.com/publications/dose.pdf):
> > >
> > > >"Many of these dataset pairings are **“simple”,** in that likelihood alone would be a reasonable rule to detect OOD data. However, there are several **“hard”** OOD dataset pairings identified by previous work. FashionMNIST→MNIST and CIFAR10→SVHN were both identified as difficult dataset pairings by Nalisnick et al. [2019a]. Additionally, Nalisnick et al. [2019b] identified CelebA→ CIFAR10/100 and CIFAR10→CIFAR100 to be particularly difficult pairings. The latter is particularly difficult, since both are subsets of the 80 million tiny images dataset [Torralba et al., 2008], but have non-overlapping class labels."
> > >
> > > Thus, as you especially emphasize the "harder" task, we **wrongly followed DoSE authors' definition** of it and added the experiments on "CelebA(ID) / CIFARs(OOD)". Please allow us to express our apologies again.
> > >
> > > For comparisons on "VFlip and HFlip", we pretty much appreciate your thoughtful comments "OOD differs from IID by only one latent factor", which could be an interesting perspective for evaluating the PHP method. In this case, the DEC method is totally useless since the calibration term $\mathcal{C}(x)$ of a data sample is the same as its flipped version, and AVOID's performance will be the same as the PHP's performance. However, we can only partly agree with this about the "VFlip", where **"HFlip" seems to be very weird and meaningless in some datasets**. Take the CelebA dataset for example, how can we know whether an unseen testing sample (a "face") is horizontally flipped or not? Therefore, we found that our PHP method could generally improve the OOD detection performance on the "VFlip" experiments but no significant improvement on the "HFlip" experiments as shown below.
> > >
> > >
> > >
> > > | AUROC of "VFlip"        | CelebA | CIFAR10 | SVHN  | FashionMNIST | MNIST  |
> > > | ------------- | ------ | ------- | ----- | ------------ | ------ |
> > > | ELBO of VAE [25]     | 74.2   | 49.5    | 50.4  | 69.5 | 82.7   |
> > > | PHP (=AVOID)         | 85.7   | 53.7    | 52.7  | 86.2 | 84.9   |
> > >
> > > | AUROC of "HFlip"        | CelebA | CIFAR10 | SVHN  | FashionMNIST | MNIST  |
> > > | ------------- | ------ | ------- | ----- | ------------ | ------ |
> > > | ELBO of VAE [25]     | 49.6   | 50.5    | 50.6  | 68.4 | 83.4   |
> > > | PHP (=AVOID)         | 50.1   | 50.4    | 50.5  | 70.2 | 85.3   |
> > >
> > >
> > > The experiments on "VFlip" could support our analysis of the latent factor and the effectiveness of the PHP method.
> > >
> > >
> > > We sincerely value your insightful feedback, as it could substantially enhance the quality of our paper. We hope our response could adequately address your concerns and if there are any remaining concerns, we are more than willing to engage in further discussion with you.

---

> > > ### Author Response · Authors · 2023-08-18
> > >
> > > Dear reviewer ogin,
> > >
> > > We greatly appreciate your continued engagement in the discussion. We have included your mentioned "hard" tasks in our secondary response. We sincerely hope these experimental results can address your concerns and demonstrate the effectiveness of our developed method.
> > > If there are any remaining concerns, we are more than willing to engage in further discussion with you.
> > >
> > > Best regards,
> > >
> > > Authors

---

> > > > ### Comment · Reviewer_ogin · 2023-08-21
> > > > **Sorry for the late response and thank you for the efforts**
> > > >
> > > > Your results and explantions address most of my concerns. However, I respectfully disagree with the claim:
> > > >
> > > > > However, the influence of this term will become more and more slight with the introduction of more powerful neural networks and optimization algorithms, and it is also beyond the investigation scope of this paper.
> > > >
> > > > I encourage the authors to read into [1] [2]. This problem may not be solvable for a long time within VAE. Moreover, my comments are independent of model training, whether your method considers it or not. It is about VAE's inherient model mis-specification. Again, [1] [2] can bring more insights. The experiments nonetheless, show your method does work improve in the harder cases, and the value of the theoretical calculation is supported. I'm raising my score to 5.

---

> > > > > ### Author Response · Authors · 2023-08-21
> > > > >
> > > > > Thanks for your response and also your recognition of our work. We think we have got the main idea of your comments and will carefully investigate the insights proposed in these papers [1][2]. Due to the limited time, we promise that we will try to discuss the model mis-specification of VAE in our paper and also how to alleviate this issue from the perspective of our methods.

---

### Author Rebuttal · Authors · 2023-08-10

**For all reviewers: Introduction to additional experiments in the attached "one-page rebuttal pdf"**

First of all, we would like to extend our sincere gratitude to all the reviewers for their meticulous reviews, thoughtful comments, and valuable suggestions. Their feedback has greatly contributed to the improvement and clarity of this manuscript. We deeply appreciate the time and effort invested in guiding our work.

We've conducted the following experiments in response to the feedback from the reviewers:

**1. Evaluating proposed methods on "harder tasks":**
As suggested by **reviewer ogin**, we add comparisons between our methods and other VAE-based OOD detection methods on the "harder tasks", i.e., detecting CIFAR-10/100 as OOD with VAEs trained on CelebA (ID).  The experimental setup remains consistent with that detailed in Section 5.1 of the original paper. The results are presented in **Table 1** of the attached one-page rebuttal pdf file. As indicated in Table 1,  our methods could still effectively alleviate the overestimation and improve OOD detection performance in these harder tasks.

**2. Evaluating proposed methods on more OOD datasets with VAEs trained on CelebA:**
Following the above experiment, we testify the proposed methods' OOD detection performance on more OOD datasets in **Table 2** of the attached one-page rebuttal pdf file. The results demonstrate that our methods could generally alleviate the overestimation and improve the OOD detection performance.

**3. Exploring the effects of dataset size and model capacity in alleviating overestimation:**
In response to feedback from reviewers **LEk9** and **GLsZ**, we investigated the influence of dataset size (amount of training data) and model capacity (number of neural network layers) on the OOD detection performance of ELBO, using both the synthesized 2D multi-modal dataset and realistic image datasets  ("FashionMNIST(ID) / MNIST(OOD)" and "CIFAR-10(ID) / SVHN(OOD)"). Our findings are illustrated in **Figure 1 (a-b)** and **Table 3** of the attached one-page rebuttal PDF.
For the 2D multi-modal dataset, we sampled a data volume 10 times greater than its inherent distribution
p(x) than the original configuration seen in Figure 3(a-b) of the main paper, increasing from 10,000 to 100,000 training samples. The VAE for this experiment utilized a 10-layer MLP as opposed to the original 3-layer MLP. Notably, the results from Figure 1(a) highlight that the $q_{id}(z)$ is still not equal to p(z) = N (0, I) and Figure 1 (b) indicates indicates the persistence of the overestimation problem in the non-linear deep VAE. For the practical image datasets, we varied the dataset size and model capacity (number of CNN layers) to investigate their effects on ELBO's OOD detection performance.
However, results show that increasing the amount of data and the number of CNN layers does not yield significant improvements.

**4. Training curve:**
In response to the concern raised by reviewer **Bzqi** regarding the number of training epochs, we have illustrated the training curve of the negative ELBO in **Figure 1 \(c\)** of the attached one-page rebuttal pdf file, based on a VAE trained on the CIFAR-10 dataset. These results are drawn from five random runs with distinct seeds. The negative ELBO rapidly decreases within the initial 200 epochs and subsequently stabilizes in the following epochs.

**5. ROC curve and corresponding AUROC value:**
In addressing the concerns of reviewer **LEk9** about the AUROC value for PHP, we've depicted the ROC curve for PHP, AVOID, and ELBO using the "FashionMNIST(ID) / MNIST(OOD)" dataset pair. This can be viewed in **Figure 1(d)** of the attached one-page rebuttal PDF.

Additionally, we've included a **discussion addressing the frequently raised concerns about the limitations of our work**:
Identifying the underlying root cause in all DGM methods which suffer from the overestimation issue could be very challenging now, since the training paradigms and model architectures vary significantly across DGMs, e.g., the training objective of flow models is the exact marginal likelihood but VAE's training objective is an evidence lower bound (ELBO). But this direction is very attractive and we would keep on researching it.

---

### Author Response · Authors · 2023-08-21
**Summary of the rebuttal by authors**

Firstly, we would like to express our sincere gratitude for the diligent efforts made by the reviewers during the discussion phase. Their valuable suggestions have significantly improved the quality of our paper! As the rebuttal content is quite extensive and lengthy, we have opted to provide a condensed summary of the key strengths of our paper that were acknowledged by the reviewers. Additionally, we have outlined certain remaining concerns raised by the reviewers that may warrant further discussion in the upcoming stages.

---
**The key strengths acknowledged by the reviewers:**

**1. The theoretical analysis and methods are inspiring for the unsupervised OOD detection community:**
1) Decomposing the ELBO carefully is interesting. In particular, they give a new prior design targeting the overestimation issue.[reviewer ogin]
2) The theory of the paper is simple but inspiring and matches neatly with the designed algorithm. [reviewer Bzqi]
3) The clear definition of overestimation in Eq. (3) following is useful, also for the wider literature. [reviewer LEk9]
4) Decomposing the expected ELBO into two components is quite crucial to understand the underline benefit and drawback to use ELBO as OOD score. [reviewer GLsZ]
5) The observation well inspires the proposed post-hoc prior method. [reviewer GLsZ]

**2. Experiments are sufficient and could support the analysis:**
1) A scoring method that improves upon the standard ELBO, which partially validifies their analysis. [reviewer ogin]
2) The experiments are well-designed and executed. The ablation studies are well-done. [reviewer Bzqi]
3) The experimental setup is large: Table 3 demonstrates that the number of dataset combinations considered is numerous, in particular in comparison to previous work. [reviwer LEk9]
4) Experiment includes varied OOD detection methods including supervised, auxiliary, and unsupervised (ensemble/non-ensemble), and shows the proposed method beats baselines within a specific category. [reviewer GLsZ]

---
**The remaining concerns raised by the reviewers after the rebuttal:**

**1. Add the experiments on harder (VFlip and HFlip) cases [reviewer ogin]:** Thanks for reviewer ogin's response, this concern has been addressed.

**2. "How the conclusion in 180-183 'can be flexibly extended for other non-linear VAEs', please explain this more" [reviewer LEk9]**: We found this is a misunderstanding of the "extension" and we have added an explanation on the secondary response (https://openreview.net/forum?id=31zVEkOGYU&noteId=0SEv4Fm3Ml). Briefly, it is due to that the condition $q(z)=p(z)$ is hard or even impossible to achieve with a trained VAE on an in-distribution training set in practice, no matter what the model architecture is. Thus, the findings in part 3 and Figure 3 of section 3.2 (a specific MLP-based VAE and a specific CNN-based VAE) could be extended to "other" non-linear VAEs with various network architectures.

**3. The design of the calibration term (DEC method) is yet not well motivated and DEC seems to be highly limited to image data [reviewer LEk9]:**
Firstly, there is a big misunderstanding that we don't use image complexity to estimate dataset entropy in the original paper. Then, for the motivation, the DEC method is applied to offset the negative impact caused by the gap of dataset entropies, which should have two properties: $\mathbb{E}\_{id}[\mathcal{C}(x_{id})]>\mathbb{E}\_{ood}[\mathcal{C}(x_{ood})]$ and $\mathbb{E}\_{id}[\mathcal{C}(x_{id})]\approx\mathcal{H}\_{id}$. Note that, $\mathcal{H}\_{id}$ is estimated by the PHP method, i.e., $\mathcal{H}\_{id}\approx\mathbb{E}\_{id}[PHP(x)]$, instead of the image complexity. As this is also related to a concern of **reviewer GLsZ**, we also added a VFlip example to demonstrate it (https://openreview.net/forum?id=31zVEkOGYU&noteId=QSTNNYppKw).
We feel very sorry that the organization of the paper caused this misunderstanding although we have already explained it in sections 4.2 and 4.3 of the main paper for how we design the DEC method, i.e., the motivation. The experimental results of the DEC method could also support the effectiveness of the design in alleviating overestimation.

For "limited to image data",  we responded at https://openreview.net/forum?id=31zVEkOGYU&noteId=0SEv4Fm3Ml. Briefly, the analysis is not limited to the data type. The choice of the SVD-based DEC method is due to that the common benchmarks of OOD detection are on image datasets. Besides, SVD could be applied to more general "matrix" and high-dimensional data.

---
It is a huge pity that we cannot know whether our responses could address reviewers' remaining concerns until now due to the limited time of the rebuttal. We sincerely hope that this brief summary can clarify reviews' concerns during the next stage of discussion.

---

### Decision · Program_Chairs · 2023-09-21

**Decision:**

Reject

**Comment:**

The paper uses decomposes the ELBO at an optimality criteria into two terms to analyze the overestimation problem of out-of-distribution data relative to in-distribution data for out-of-distribution detection in VAEs. They attribute this overestimation to choices in the prior distribution and a difference of entropies. The develop two techniques based on this approach a post-hoc prior and recalibration for entropies. The average scores for this paper sit right at borderline accept. There were questions about the reasonableness of the optimality assumptions for a trained VAE and whether they are actually realized. There were also a harder collection of tasks that were requested. Part of the challenge for this paper was in writing and in the quality of the existing experimental results. The most negative reviewer states that with the new results the experiments are improved.

There are also alternative explanations for the failure of out of distribution detection even outside of VAEs that are worth discussing and integrating into the paper. See for example [Understanding failures in out-of-distribution detection with deep generative models](http://proceedings.mlr.press/v139/zhang21g/zhang21g.pdf) by Zhang et al. in ICML 2021.

Overall, this paper seems to me as borderline, but one that would benefit from integration of the reviewer feedback and review of a new manuscript.